# Dielectrophoretic bead-droplet reactor for solid-phase synthesis

Punnag Padhy [1] ✉, Mohammad Asif Zaman[1], Michael Anthony Jensen [2,3] ✉, Yao-Te Cheng[1], Yogi Huang[4], Mo Wu [1], Ludwig Galambos[1], Ronald Wayne Davis[2,3,5] & Lambertus Hesselink[1] ✉

Solid-phase synthesis underpins many advances in synthetic and combinatorial chemistry, biology, and material science. The immobilization of a reacting species on the solid support makes interfacing of reagents an important challenge in this approach. In traditional synthesis columns, this leads to reaction errors that limit the product yield and necessitates excess consumption of the mobile reagent phase. Although droplet microfluidics can mitigate these problems, its adoption is fundamentally limited by the inability to controllably interface microbeads and reagent droplets. Here, we introduce Dielectrophoretic Bead-Droplet Reactor as a physical method to implement solid-phase synthesis on individual functionalized microbeads by encapsulating and ejecting them from microdroplets by tuning the supply voltage. Proof-of-concept demonstration of the enzymatic coupling of fluorescently labeled nucleotides onto the bead using this reactor yielded a 3.2-fold higher fidelity over columns through precise interfacing of individual microreactors and beads. Our work combines microparticle manipulation and droplet microfluidics to address a long-standing problem in solid-phase synthesis with potentially wide-ranging implications.

Solid-phase synthesis techniques[1–3] are indispensable for the de novo synthesis of oligonucleotides (single stranded DNA[4–6] and RNA[7]), oligopeptides[1,3,8], oligosaccharides[9], and combinatorial libraries[10]. In this approach, long chained polymeric molecules are synthesized by stitching one monomeric unit at a time onto initiator (primer) strands that are bound to insoluble solid supports[1,8,11,12]. Immobilization of one reacting species on the solid supports brings forth the challenge of adequate reagent interfacing in solid-phase synthesis. The resultant incomplete reactions lead to accumulating deletion errors ($\xi$) over multiple synthesis cycles that decrease the product purity and yield ($\eta = (1 - \xi)^N$)[4–6,13,14] with increasing strand length (larger $N$) of the polymeric chain. In general, this is addressed by flushing the solid supports with a large excess of mobile reagents in the fluidic phase in traditional synthesis columns[1,8,11,12]. However, this entails significant

reagent consumption and wastage[1,6,8,11,12,15,16]. Hence, minimizing reagent consumption and wastage as well as reaction errors ($\xi$) is necessary to realize the scale of synthesis effort required to engineer complex molecules through an iterative design-build-test-learn approach[5,15,17].

Droplet microfluidics[18–25] is a potent platform that can address these issues. The small size of microdroplet reactors ($1\,\mu m - 500\,\mu m$ in diameter) can minimize reagent wastage[19,21,23,25]. It can enable superior interfacing of mobile reagents with solid supports due to faster mixing (faster diffusion times in smaller droplets) of reacting species[18–20,22,25–27]. Furthermore, due to the advent of on-demand droplet generation, a single droplet can be dispensed into the reaction chamber at rates determined by the synthesis and detection process. Unlike the continuous stream droplet generation, this prevents the

[1]Department of Electrical Engineering, Stanford University, Stanford, CA 94305, USA. [2]Stanford Genome Technology Center, Stanford University, Palo Alto, CA 94304, USA. [3]Department of Biochemistry, Stanford University, Stanford, CA 94305, USA. [4]Department of Chemical Engineering, Stanford University, Stanford, CA 94305, USA. [5]Department of Genetics, Stanford University, Stanford, CA 94305, USA. ✉e-mail: punnag@stanford.edu; m.a.jensen@stanford.edu; hesselink@ee.stanford.edu

fluid flow driving the droplet generation from impacting the chemical reaction and detection processes[25,28–32]. These advantages have attracted the attention of chemists[33,34], biologists[35–39], and material scientists[40–42] alike towards developing droplet reactors in a microfluidic platform which has led to the tremendous growth of the field in the last few decades. However, the reagents in all these instances are entirely in the fluidic phase[33–42]. Despite its many decisive advantages, droplet microfluidic analog for solid-phase synthesis, though highly desirable[15,43], remains an unmet scientific and engineering challenge due to the inability to controllably interface solid supports with reagent droplets in a droplet microfluidic platform.

Here we combine dielectrophoresis[44–50] with on-demand droplet generation[25,28–32] and state-of-the-art device fabrication to introduce Dielectrophoretic Bead-Droplet Reactor (DBDR). This is an approach to implement solid-phase synthesis reactions in droplet microfluidic systems by controllably encapsulating and ejecting (interfacing) individual microbeads (of radius $R_b$) from single reagent microdroplets (of radius $R_d$). The capillary force of interfacial tension[45,51,52] ($\gamma_{ow}$), which is significant at this length scale, impedes the bidirectional motion of the bead across the interface. It drives the bead towards the interfacial energy minimum, the position of which is governed by the intrinsic interfacial properties of the bead, droplet and the medium[45,51,52]. Therefore, an external force is required to manipulate the bead entirely into and out of the droplet across the droplet-medium interface by overcoming the capillary force. Micromanipulation techniques[53,54] such as dielectrophoretic, magnetic, optical, and acoustic tweezers, in which forces scale with the particle volume ($\propto R_b^3$), have traditionally been employed to maneuver microparticles in a single fluidic suspension medium against the viscous drag force ($\propto R_b v$) at a velocity $v$ relative to the medium[45]. However, manipulation of microparticles across micron scale immiscible fluidic interfaces remains challenging due to the large capillary force. It scales linearly with the particle size[45,51,52] ($\propto R_b$) compared to the volume scaling ($\propto R_b^3$) of the micromanipulation body forces[53,54]. Here we show that by judicious design of the interfacial, and geometrical properties of the bead, droplet, and the suspension medium the balance between the counteracting dielectrophoretic and capillary forces can be modulated by simply tuning the supply voltage ($V_s$) to completely encapsulate and eject the bead out of the droplet (Fig. 1a) in a droplet microfluidic device (Fig. 1b, c). Specifically, we show that when sufficiently large voltage ($V_s$) is applied across the electrodes, the dielectrophoretic force on the droplet ($\propto R_d^3 |V_s|^2$) overcomes the capillary forces ($\propto R_b \gamma_{ow}$) to move the hydrophobic bead into the droplet (across the aqueous reagent-oil medium interface). At low voltages ($V_s$), the dominant capillary force ejects the hydrophobic bead out of the droplet. In all prior reports of microparticle encapsulation, beads presuspended in the dispersed (droplet generating) phase are either engulfed within the droplet during its formation[55,56] or are injected into another droplet[57]. In these approaches, once encapsulated, the beads remain confined within the droplet either by the capillary force (hydrophilic beads) or by acoustic forces which push the beads to the center of the droplet (away from the droplet-medium interface). In all these processes, the bead does not move across an interface of immiscible fluids. While external fields and microchannel architectures may be used to trigger droplet breakup[57], the beads do not move across any fluidic interface. On the contrary, by employing counteracting capillary and dielectrophoretic force, DBDR ensures bidirectional manipulation of the bead across the droplet-medium interface for its encapsulation within the droplet (and thus interfacing with the reagents) as well as its ejection from it (and thus separation from the reagents). Ejection will allow the fluorescent characterization of the products bound to the beads after a reaction step.

We use DBDR to make a proof-of-concept demonstration of the enzymatic coupling of fluorescently labeled nucleotides to the 3′ end of the initiator strands bound to the bead surface (Fig. 1d).

Furthermore, fluorescence-based comparisons of synthesis reactions implemented using DBDR with synthesis columns indicate improved coupling fidelity due to superior interfacing of beads with reagents.

## Results

### Device and reaction design

The silicon-on-glass microfluidic device developed for demonstration of DBDR consists of Indium Tin Oxide (ITO) electrodes ($\approx 15\,\mu m$ wide, Fig. 1b, c and Supplementary Fig. 1) in the reaction chamber ($3\,mm \times 10\,mm \times 0.2\,mm$). These electrodes are suitably aligned with the on-demand droplet generator microchannel[34,37–41] ($\approx 25\,\mu m$ wide and $\approx 20\,\mu m$ height, Fig. 1b, c and Supplementary Fig. 1) to ensure that the droplet ($R_d \approx 25\,\mu m$) dispensed into the reaction chamber lies within their trapping range. A piezo driven pressure controller is used to drive fluid flow through the microchannel. The abrupt release of microchannel confinement at the intersection with the reaction chamber leads to sharp change in capillary pressure which triggers the droplet breakup[31]. This avoids the impact of flow rate variations (due to the external noise from fluidic controller hardware) on the droplet size (which depends solely on the device geometry) to generate monodisperse droplets. The electrodes are made smaller than the channel (Fig. 1c) to ensure that the trapped droplet encapsulates the trapped bead as it covers the electrode (Fig. 1a). ITO and glass ensure optical transparency for imaging the trapping and manipulation process as well as fluorescence detection of the coupling reaction (Fig. 1c).

Streptavidin-coated fluorescent green polystyrene beads ($R_b = 3\,\mu m$, Fig. 1d) are used as solid supports. 5′ biotinylated oligos with 25 nucleotides bound to the microbead using streptavidin-biotin linkages (Fig. 1a, d(i)) serve as initiators. The enzyme terminal deoxynucleotidyl transferase (TdT)[58–62] couples the nucleotides fluorescently labeled with Alexa Fluor 647 (dCTP-AF647) to the 3′ end of the initiators (Fig. 1d(ii)) when the bead is encapsulated within the reagent droplet (Fig. 1a). A red LED excites AF647 to detect the coupling of dCTP-AF647 to the initiated bead[63,64] (Fig. 1a, d(ii), Supplementary Fig. 2).

### Physical working principle of DBDR

The device, filled with $\approx 2.5\%$ w/w solution of Span 80 in silicone oil (kinematic viscosity = 1 cSt), is mounted on the sample holder with relevant electrical and fluidic connections (Supplementary Fig. 3). Beads functionalized with appropriate initiator strands (Fig. 1d) are suspended in the oil solution and introduced into the device through the bottom fluidic port (Fig. 1c). When a single bead enters the vicinity of the electrodes it gets dielectrophoretically trapped (Fig. 2b) by applying a voltage ($V_s = V_{BD} = 40\,V$ amplitude) across the electrodes ($E_1$ and GND in Fig. 1c and Fig. 2b). The voltage supply is then lowered. Subsequently, a reagent droplet is dispensed into the reaction chamber (Fig. 2c) by applying a pressure pulse using a piezoelectric pressure controller (Elveflow OB1 MK3+). The magnitude ($\approx 10\,mbar$) and the duration of the pulse ($\approx 300\,ms$) are critical to generate a single droplet. Then the voltage supply is switched back on ($V_s = V_{DD} = 20\,V$) to trap the droplet adjacent to the bead (Fig. 2d).

When $V_s$ is increased to $\approx 120\,V$ at 200 Hz supply frequency ($f_s$), the bead is fully encapsulated by the reagent droplet (Fig. 2e(i), Supplementary Movie 1). Reducing $V_s$ to $\approx 0.1\,V$ ejects the bead from the droplet (Fig. 2e(i), Supplementary Movie 1). Electrohydrodynamic simulations (Fig. 2e(ii), details in Supplementary Notes 1–4, Supplementary Figs. 4–6, and Supplementary Table 1) support these observations. It can be understood in terms of the change in the system's electrocapillary potential energy ($\triangle U$)[51,52] due to a change in the Gibbs free energy ($\Delta U_{IT} \approx -\gamma_{ow} \cos\theta \triangle A_{ws}$) of the interfaces between the bead, droplet and the medium as well as the change in the electrical energy ($\Delta U_E = -\Delta(QV_s)/2$) stored in the system (Fig. 2e(iii), (iv)). Here, $\gamma_{ow} = 5.5\,mN/m$, $A_{ws}$ is the area of the bead covered by the reagent droplet, $\theta \approx 145°$ is the contact angle that the reagent droplet forms

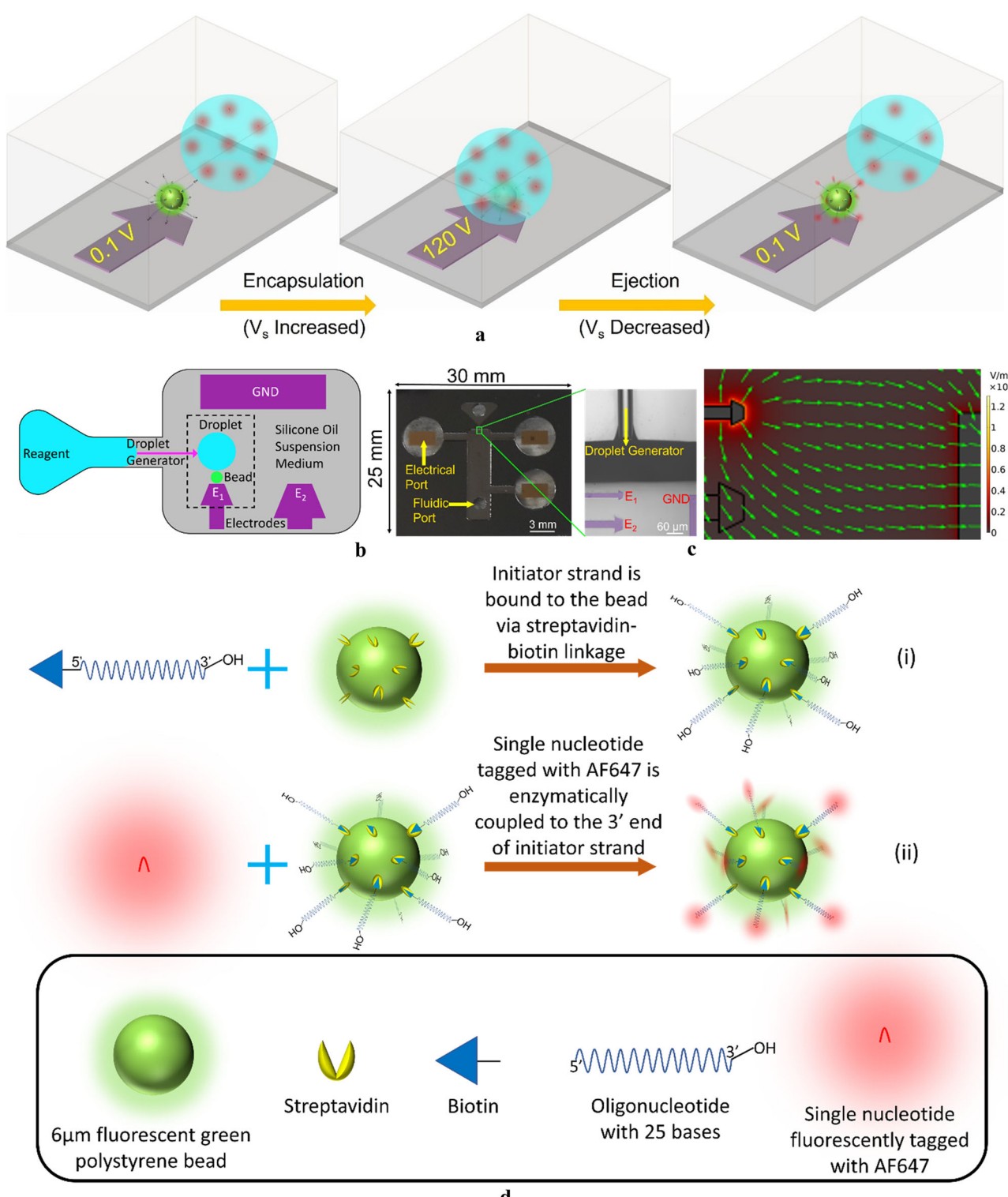

**Fig. 1 | Conceptual representation of proposed Dielectrophoretic Bead-Droplet Reactor (DBDR). a** In DBDR reactions are implemented by encapsulating and ejecting individual beads from reagent droplets. This is achieved by simply tuning the voltage supply on the trap electrode that modulates the balance between counteracting dielectrophoretic and capillary force. Inside the droplet, the fluorescently labeled nucleotides (dCTP-AF647) couple onto the initiator strands on the bead. **b** The device footprint depicts the alignment of the droplet generator and the electrodes in the reaction chamber and the spatial location of the bead and the dispensed droplet. The dashed box shows the section of the device in (**a**).

**c** Fabricated silicon-on-glass microfluidic device with ITO electrodes. Purple color is overlaid on the transparent ITO electrodes to distinguish them from the device background. The strong electric field gradient of the electrodes enables dielectrophoretic trapping. The arrows indicate the direction of the electric field.
**d** Representation of chemical reactions on the solid support: (i) Binding the initiator strand on the bead through streptavidin-biotin hydrogen bonds, (ii) Enzymatic coupling of fluorescently labeled nucleotides (dCTP-AF647) onto the 3' end of the initiator strands on the bead inside the reagent droplet.

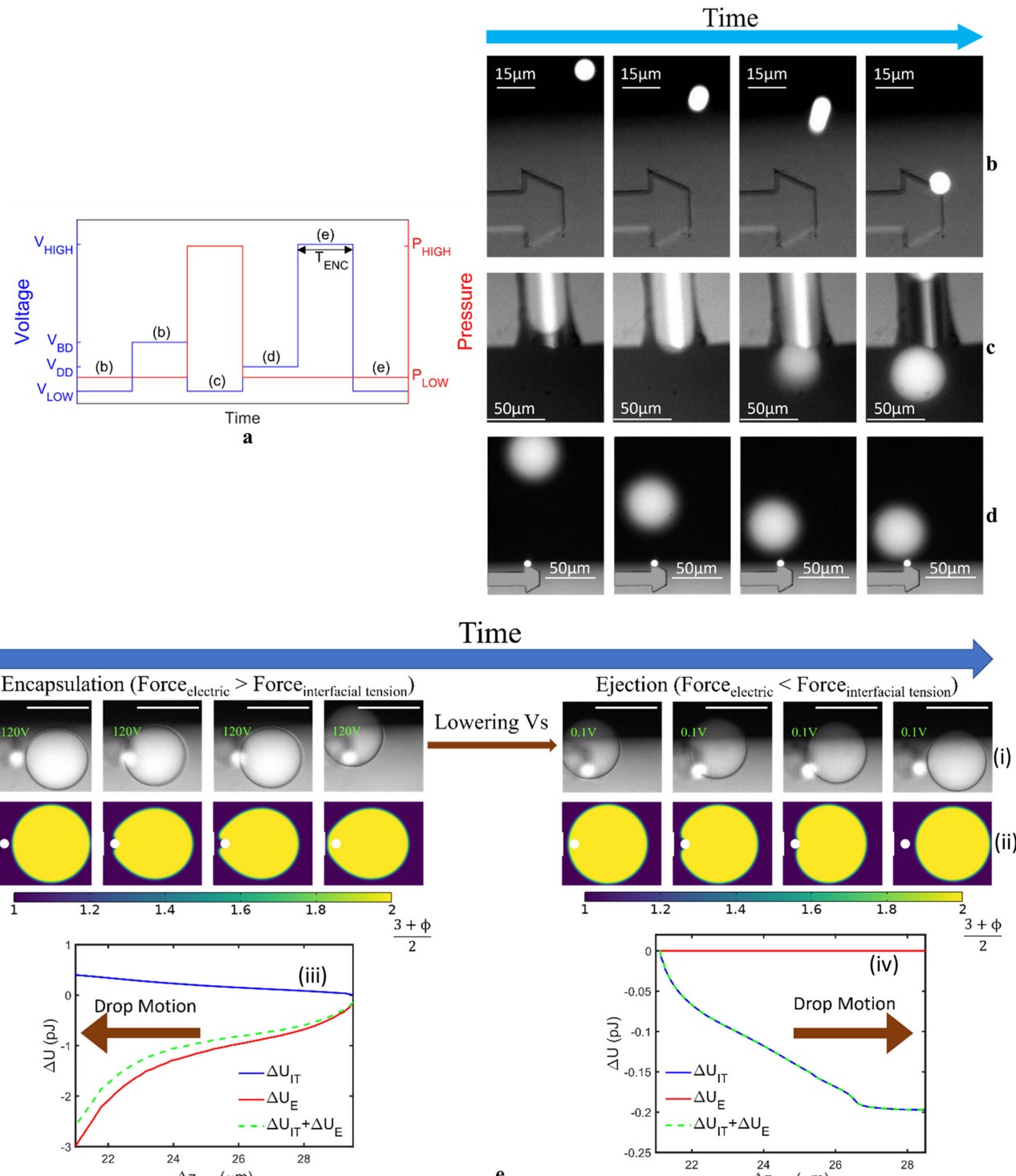

**Fig. 2 | Experimental demonstration and numerical simulation of the physical processes underlying DBDR. a** Schematic plot of the sequential change in the supply voltage and pressure required to drive the physical process underlying DBDR. The voltage supply is used to manipulate the bead and the droplet in the reaction chamber. The pressure pulse dispenses a droplet on-demand into the reaction chamber. $T_{ENC}$ represents the time for which the bead remains encapsulated within the droplet. **b** A single bead in the vicinity of the electrode is dielectrophoretically trapped by raising the voltage to $V_S \approx 40$ V. **c** The voltage is lowered, and a pressure pulse is exerted on the microchannel to dispense a single droplet into the reaction chamber. **d** The pressure pulse is stopped and a voltage $V_{DD} \approx 20$ V is applied to trap the generated droplet adjacent to the trapped bead. **e** (i) Experimental implementation and (ii) numerical electrohydrodynamic phase field simulation depicts the encapsulation of the bead by the droplet under high supply voltage ($V_s \approx 120$ V) and its ejection from the droplet under low supply voltage ($V_s \approx 0.1$ V). The phase field variable (φ) has a value of −1 in the silicone oil suspension medium (phase 1) and a value of 1 in the reagent droplet (phase 2). φ transitions from −1 to 1 at the droplet-medium interface. The bead is represented in white. (iii) Electrocapillary potential energy representation of the engulfing and (iv) ejection process. The frames in e(i) are extracted from Supplementary Movie 1 which was recorded at 30 fps. The scale bars in these frames represent 50 μm.

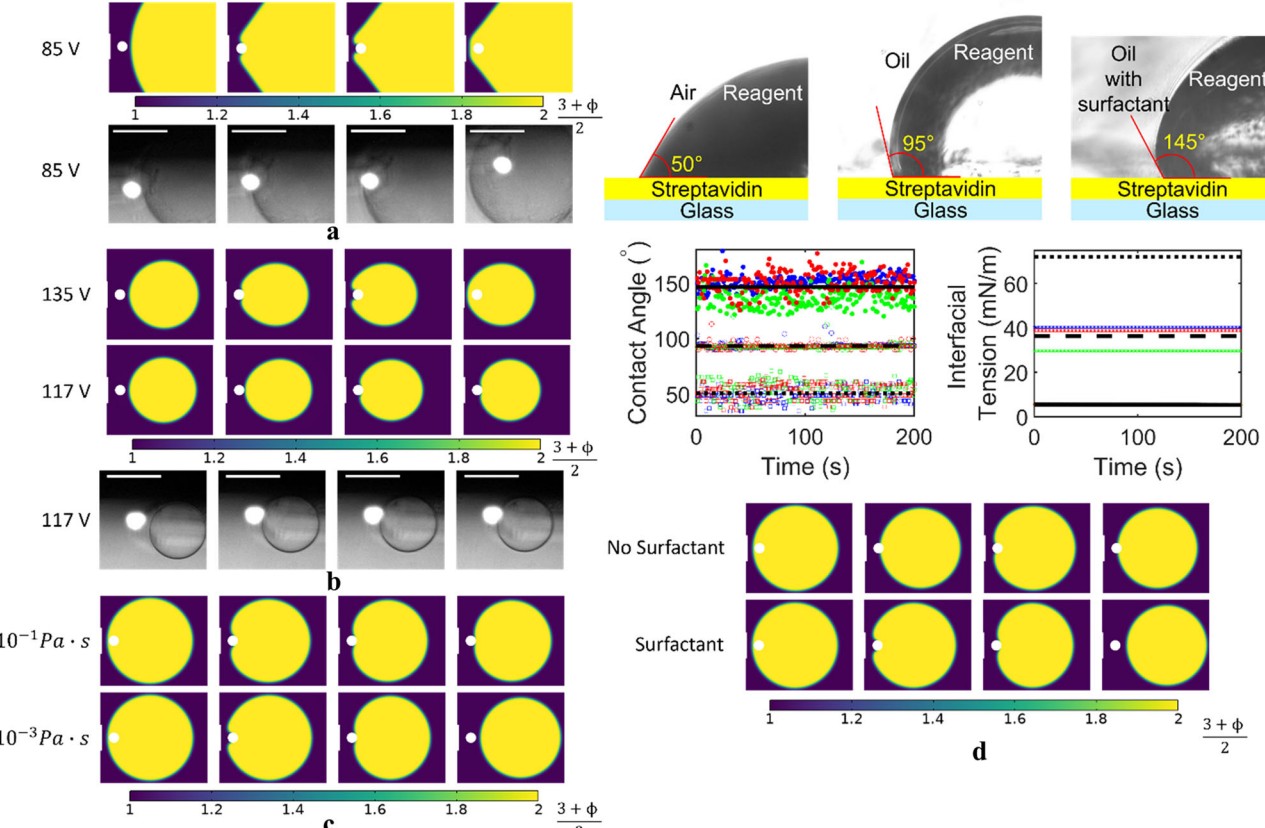

**Fig. 3 | System design for encapsulation and ejection. a** Larger droplet ($R_d = 50\,\mu m$) can encapsulate the bead at a much lesser supply voltage ($V_s \approx 85V$). **b** Smaller droplet ($R_d = 20\,\mu m$) requires a larger supply voltage ($V_s \approx 135V$) to overcome the capillary force and encapsulate the bead. At the lower voltage ($V_s \approx 117$ V) the dielectrophoretic force cannot overcome the capillary force to encapsulate the bead into the smaller droplet (It was enough to encapsulate the bead in the $R_d = 25\,\mu m$ droplet). **c** The low viscosity of silicone oil 1 cSt enables the ejection of the bead from the droplet at $V_s \approx 0.1$ V. With increase in oil viscosity the increased dissipation of the kinetic energy of the droplet prevents complete separation from the bead. **d** Silicone oil 1 cSt (dashed line) renders the hydrophilic (in air, dotted line) streptavidin surface slightly hydrophobic. The surfactant

Span80 further reduces the interfacial tension between the silicone oil and the reagent and increases the contact angle (solid line) that the reagent droplet forms on a streptavidin surface making it hydrophobic. It is critical for the ejection of the bead from the droplet. Interfacial tension and contact angle measurements are recorded with time variations to account for the surface adsorption. (Refer to Experimental Procedure for Bead-Droplet Interaction subsection of Methods for details.) The streptavidin and glass thicknesses are not drawn to scale. The scale bars on experimental frames in (**a**, **b**) represent 40 $\mu$m. The symbol $\phi$ in the color plots is the phase variable in the phase field simulations. $\phi = -1$ in the oil medium (represented in blue) and $\phi = 1$ in the aqueous medium (represented in red). $-1 < \phi < 1$ represents the boundary region between the droplet and the oil medium.

on the surface of the bead and $Q$ is the charge stored on the electrode $E_1$. The large $\theta$ means the bead has a propensity towards the oil (hydrophobic). When $V_s$ is high, $|\Delta U_E| \gg |\Delta U_{IT}|$. To attain the minimum energy configuration, the droplet moves towards the electrode ($\Delta U \approx \Delta U_E < 0$) and engulfs the bead in the process (Fig. 2e(iii)). When $V_s$ is low, $|\Delta U_E| \ll |\Delta U_{IT}|$. To attain the minimum energy configuration, the droplet ejects the hydrophobic bead, which is touching the droplet-medium interface from the inside, as it moves away from the electrode ($\Delta U \approx \Delta U_{IT} < 0$, Fig. 2e(iv)). Once the bead completely separates from the droplet, $\Delta U_{IT}$ is negligible (Fig. 2e(iv)). The process of encapsulation (Supplementary Movie 2) and ejection (Supplementary Movie 3) proceed over $\approx 200 - 400$ ms. This fast transition time can ensure that the synthesis rate is not limited by the physical sample handling process. The pressure pulse for on-demand single droplet generation (Fig. 2a) was exerted when the electrical voltage was low. This ensures bead and droplet trapping as well as encapsulation and ejection proceeds without any undue effect of the fluid flow in the device or additional droplet generation. The time ($T_{ENC}$) for which the bead and the droplet remain in contact depends on the time for which the voltage is high (Supplementary Movie 1 and Supplementary Movie 4).

Many design choices are crucial towards the successful working of the above process. While a larger droplet ($R_d = 50\,\mu m$) requires a lesser

voltage ($V_s \approx 85$ V) to exert significant dielectrophoretic force ($\propto R_d^3 |V_s|^2$) to encapsulate the bead (shown in Fig. 3a) it would also mean more reagent usage per reaction and longer diffusion times. On the other hand, a smaller droplet ($R_d = 20\,\mu m$) would require a larger voltage ($V_s \approx 135$ V) to exert significant dielectrophoretic force ($\propto R_d^3 |V_s|^2$) to move the bead across the interface (Fig. 3b). In fact, this emphasizes our choice to drive the encapsulation and ejection process by exerting dielectrophoretic force on the droplet. It will experience a much larger dielectrophoretic force than the bead due to its larger size and larger permittivity contrast with the suspension medium ($\frac{F_{DEP,droplet}}{F_{DEP,bead}} \propto \left(\frac{R_d}{R_b}\right)^3 \times \left(\frac{K_{1rd}}{K_{1b}}\right)$) that can help overcome the large capillary force. Here, $K_1$ is the well-known Clausius-Mossotti factor (Supplementary Note 2 and Supplementary Fig. 5). This requires the droplet to experience a non-zero dielectrophoretic force when not encapsulating the bead (Fig. 1a, b). The alignment of the droplet generator and electrodes (Fig. 1b, c) ensures that the droplet is dispensed into the vicinity of the electrode where it can be successfully trapped by its electric force field. Once the voltage ($V_s$) is lowered, the dominant capillary force ejects the bead touching the interior surface of the droplet.

Silicone oil 1 cSt acts as a water insoluble, chemically inert, and non-ionic suspension medium with low viscosity, high dielectric

breakdown strength and low electrical conductivity that prevents the rapid evaporation of the tiny reagent microdroplets due to their high surface-to-volume ratio[29,45] (detailed discussion in Supplementary Note 4). Low viscosity of silicone oil 1 cSt minimizes viscous dissipation of the kinetic energy[45] of the moving droplet as it moves away from the electrode (towards the minimum Gibbs free energy configuration). This ensures complete ejection of the bead (Fig. 3c). Its high dielectric breakdown electric field strength ($\approx 13.8$ V/$\mu$m) allows to exert a large enough voltage across the electrodes to ensure encapsulation of the bead within the droplet without causing dielectric breakdown of the suspension medium. Its low electrical conductivity ($\sigma_o \approx 10^{-14}$ S/m) minimizes spatially non-uniform joule heating ($\propto \sigma_o |E|^2$) to limit unwanted electrothermal flow of the suspension medium[65,66]. Its low ionic concentration also limits electro-osmotic fluid flow[66]. The latter two factors ensure that dielectrophoresis is the dominant electric field effect and prevent unwanted motion of the bead and droplet under electric field driven fluid flow that could impede the encapsulation and ejection process. If oils with identical physical properties except for higher viscosities were to be used, the higher dissipation of kinetic energy would prevent complete ejection of the bead from the droplet.

Addition of the surfactant (Span80) to silicone oil 1 cSt further reduces $\gamma_{ow}$ to 5.5 mN/m and increases the contact angle ($\theta$) that the aqueous droplet forms on the streptavidin coated surface of the bead to 145° (Fig. 3d and Sample Preparation subsection of Methods). This makes the otherwise hydrophilic streptavidin surface of the bead (water droplet forms a contact angle of 50° on a streptavidin surface in air) hydrophobic. Therefore, the choice of silicone oil and surfactant is critical for the spontaneous ejection of the bead from the droplet on touching its interior surface in the absence of an applied voltage. The surfactant also increases the contact angle ($\beta$) of the aqueous reagent droplet on the silanized interior surface of the device to $\approx 138°$ (Experimental Procedure for Bead-Droplet Interaction subsection in Methods). This ensures that the device walls are predominantly wet by the suspension medium, and the aqueous reagent phase does not stick to the device walls. This is necessary for droplet generation (Fig. 2c) and its subsequent dielectrophoretic manipulation (Fig. 2d, e). The non-wetting of the device walls by the droplet phase also avoids reactor fouling (evident by the absence of any significant remnant fluorescence trail along the path of motion of the droplet phase in Fig. 2c, d, e). This would be critical to detect reactions without signal interference from unwanted sources.

## Enzymatic coupling of nucleotide in DBDR

After establishing the physical working principle of DBDR, it was used for a proof-of-concept demonstration of enzymatic coupling of nucleotides onto the initiator strands bound to a microbead. A droplet size of $R_d = 25$ μm was chosen (through the design of microchannel dimensions, Fig. 1b, c) to minimize reagent usage and their diffusion times (associated with larger droplets) while still ensuring dielectrophoretic encapsulation of the bead (increasingly difficult with smaller droplets). The droplet has red fluorescently tagged nucleotides (dCTP-AF647) at a 5 μM concentration. Therefore, the bead with $\approx 100$ attomoles of initiator strands on its surface has access to $\approx 325$ attomoles nucleotides within the volume of the droplet (forms a contact angle $\beta \approx 138°$ with the device surface, Experimental Procedure for Bead-Droplet Interaction subsection in Methods) which is[51,52] $\frac{(2-3\cos\beta+\cos^3\beta)}{3}\pi R_d^3 = 65$ pl (A slightly excess quantity of reagent was chosen to ensure ample supply of nucleotides to the initiators subject to variations arising due to sample preparations and device fabrication). The encapsulation and ejection process were triplicated along with fluorescent imaging of the bead immediately before encapsulating and after ejecting it from the droplet (Fig. 4a, Experimental Procedure for Chemical Coupling subsection in Methods and Supplementary Fig. 7a–f). While there was no red fluorescence from the bead prior to encapsulation, we observed a clear red fluorescence

from it after ejection. This confirmed the binding of the fluorescently tagged nucleotides (dCTP-AF647) to the bead surface[57,63,64]. This is a further confirmation that the bead was enveloped within the droplet when $V_s$ was high. The three repeats of this experiment (Supplementary Fig. 7g) were performed with encapsulation time of $\approx 300$ s to ensure reproducibility. To eliminate any false positives due to unincorporated nucleotides non-specifically bound to the surface of the bead and confirm chemical binding of nucleotides to the 3′ end of the initiator strands, we reiterated the above process using beads devoid of initiator strands and reagent solution without TdT (control experiment in Fig. 4b). Lack of red fluorescence from the bead after ejection confirmed the absence of non-specific binding of nucleotides to the bead surface (Fig. 4b, Experimental Procedure for Chemical Coupling subsection in Methods, and Supplementary Fig. 7h–m). This shows DBDR can be a robust approach for enzymatic DNA synthesis on individual beads in droplet reactors within droplet microfluidic devices. It brings together the disparate fields of droplet microfluidics and solid-phase synthesis.

## Coupling fidelity enhancement in DBDR

After having demonstrated enzymatic coupling of nucleotides to initiated beads using DBDR, we compared its coupling fidelity with reactions implemented in synthesis columns (Fig. 4c) in which many initiated beads are tightly packed between filters using fluorescence measurements. We observed large variations in fluorescence intensity across beads reacted in the column (Fig. 4d, Supplementary Figs. 8–10). This indicates non-uniform reagent environment in the vicinity of each bead which results in corresponding variations in the reaction fidelities. The small fraction of the beads with the brightest fluorescence closely represent the maximum attainable enzymatic coupling fidelities. The reduced fidelity on an overwhelming majority of the beads leads to an overall decrease in the average coupling efficiency (average of the histogram in Fig. 4) and suggests significant missed couplings. On the other hand, beads reacted using DBDR consistently exhibit fluorescence intensities in the vicinity of the maximum values obtained using synthesis columns. This is quantified using the average fluorescence intensity ratio ($FIR_{avg}$). $FIR_{avg} = 3.2$ in Fig. 4d indicates enhanced access of individual bead surfaces to reagents within the microdroplets which then translates to improved reaction fidelities. While excess concentration of reagents is generally used to compensate for reduced access to reagents and drive-up reaction fidelities in synthesis columns, it is also accompanied by excess reagent wastage as well[1,6,8,11–14]. This observed superior performance of DBDR compared to synthesis columns (statistical significance of the observed 3.2-fold fidelity enhancement is indicated by Welch's t test, Supplementary Table 2, and its rank transformed variant, Supplementary Table 3, described in detail in the Image Processing and Data Analysis subsection of Methods) can be attributed to the superior reagent access to the bead surfaces in DBDR than in synthesis columns where many beads are stacked together.

The stacking of beads in synthesis columns may restrict reagent access to the bead surfaces. Ideally, the beads would be stacked in regularly packed lattice arrangements[67] within the column (Supplementary Fig. 11). However, in practice, it is a rarity. Even if the beads are perfectly stacked in the columns at the beginning, on introduction of reagents into the synthesis column (say at $t = 0$ s, Fig. 5), the particles are displaced from the perfect stack due to the drag force[44] exerted by the turbulent flow[68] ($t = 0.05$ s, Fig. 5). The resultant increased interparticle spacing facilitates improved reagent access to the bead surfaces in the columns compared to the perfect stacks. Once the reagent influx stops, the particles settle down under gravity into imperfect stacks ($t = 70$ s and $t = 300$ s, Fig. 5). Different particles have different settling times. This explains the varying exposure (spread of the gray band and histogram in Fig. 4d) to reagents during the experimental

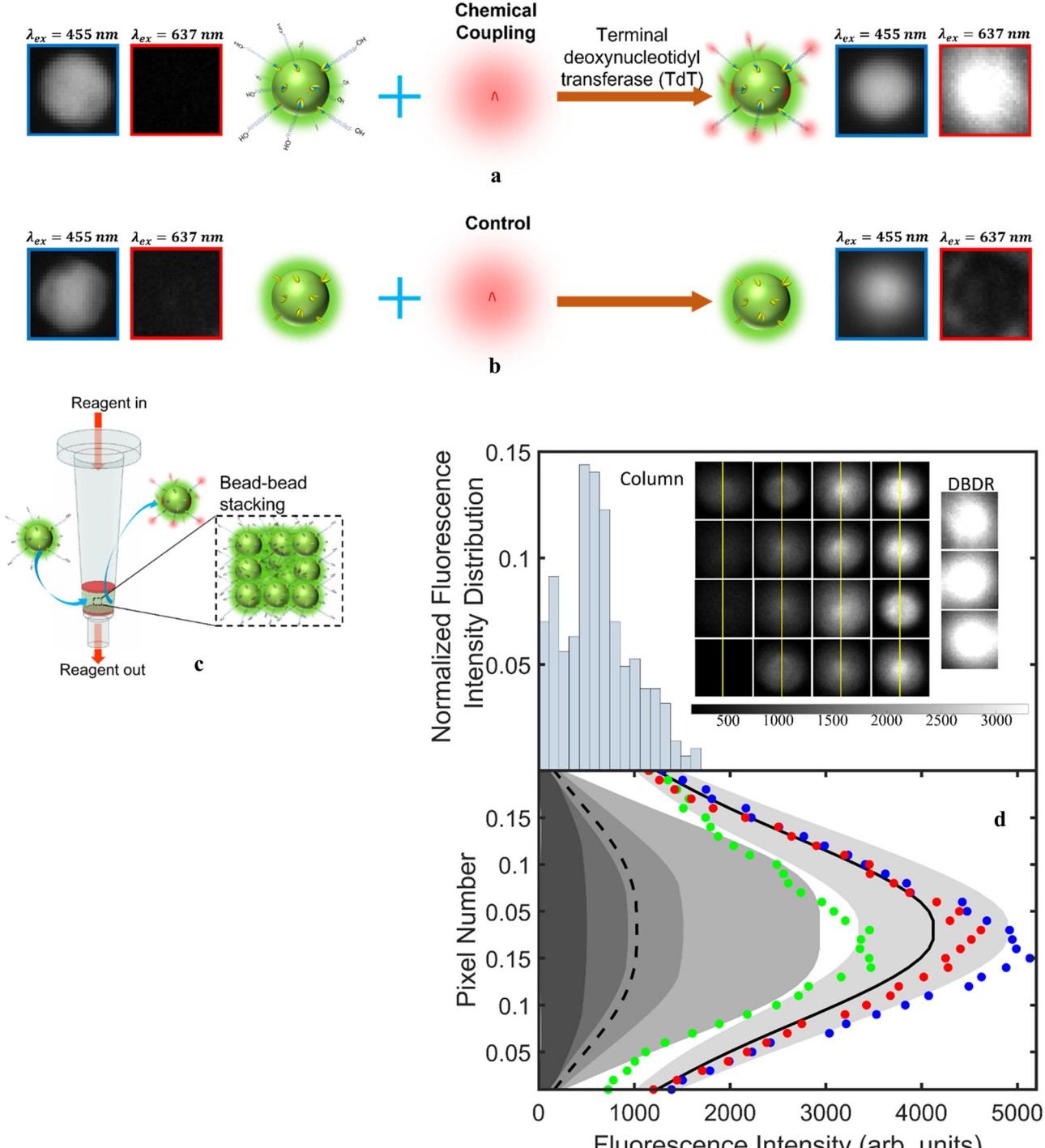

**Fig. 4 | Enzymatic coupling of nucleotides onto initiator strands on the bead and reaction fidelity analysis. a** Enzymatic coupling of fluorescently labeled nucleotides (dCTP-AF647) onto the initiator strands tethered to the green fluorescent microbead. The bead fluoresces red only after encapsulation and ejection from the droplet indicating binding of the nucleotides to its surface. **b** Control experiment was performed without initiator strands on the bead surface and enzymes in the droplet. The absence of red fluorescence from the bead after ejection from the droplet confirms that the red fluorescence from the bead in (**a**) can be attributed to the chemical coupling of the nucleotide to the initiator strands. **c** Reactions are implemented in synthesis columns by flushing reagents through the bead stack. **d** The top pane represents the average fluorescence intensity of 285 beads which reacted in synthesis columns. These fluorescences were collected across multiple frames (A few sample frames are shown in Supplementary Fig. 9a.). A few sample beads collected from all the frames are shown in the inset on the top.

The left gray band with brightness gradient in the bottom pane represents the line fluorescence intensity of these beads reacted in the column taken along the yellow lines drawn across the beads. The band edges represent the line fluorescence of beads that demarcate the quartiles of the histogram in the top pane. The dashed curve in the bottom pane represents the fluorescence intensity distribution of the bead with average fluorescence intensity equal to the mean of the histogram. The mean ($\mu_{col}$) and the standard deviation ($\sigma_{col}$) of the average fluorescence intensities of beads reacted in the synthesis column are 627.6 and 372 respectively. These are evaluated over 285 beads. The three beads which represent the three experimental trials of DBDR are shown as insets in the top pane. Their mean ($\mu_{DBDR}$) and standard deviation ($\sigma_{DBDR}$) are 2057.5 and 209 respectively. The scatter plots in the bottom pane represent their fluorescence intensities along the yellow lines. The mean and standard deviation of the three distributions is represented by the solid black curve and the right most gray band.

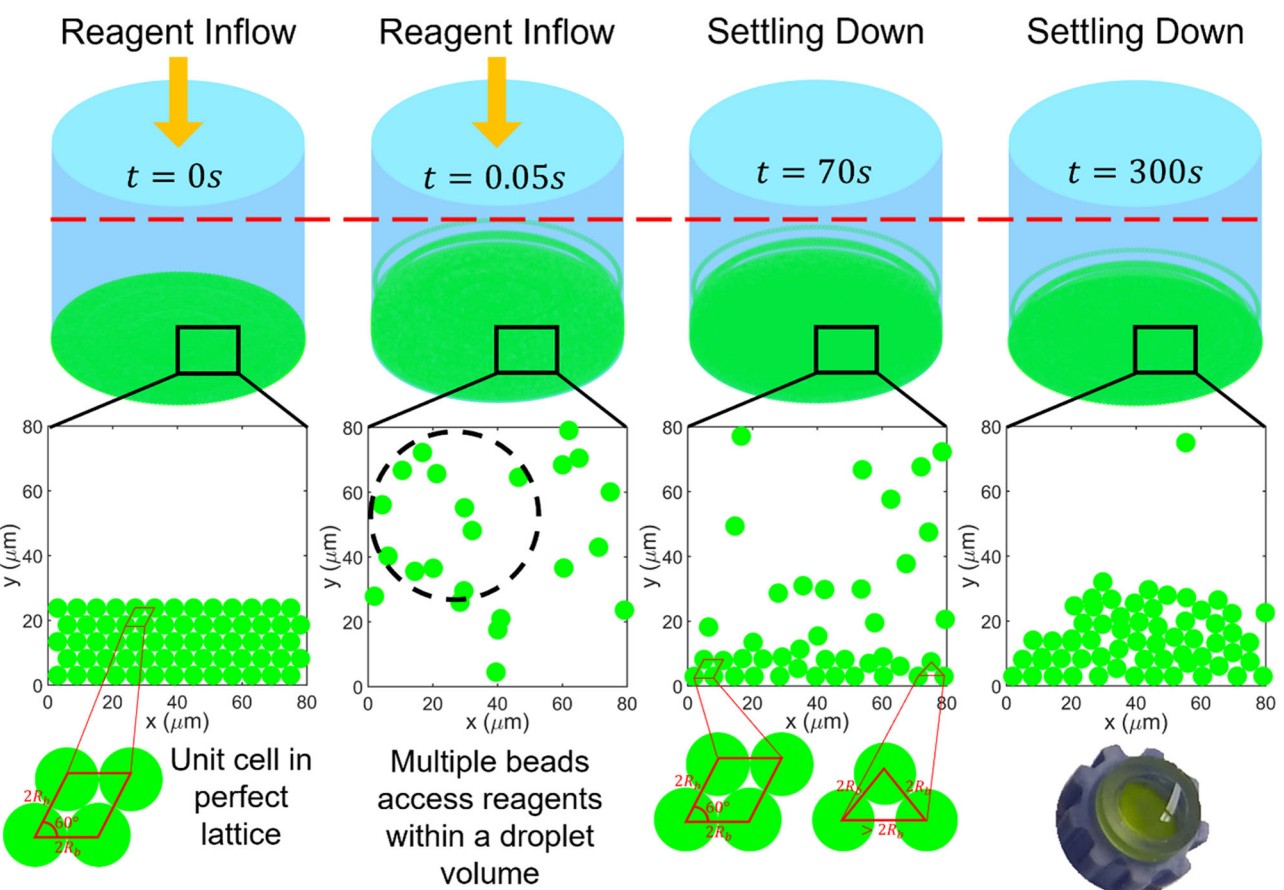

**Fig. 5 | Explaining enhanced fidelity of reactions in DBDR compared to columns.** Stacking of beads prevents reagent access to reaction sites on the bead surface in synthesis columns. Assuming an ideal scenario wherein beads were initially stacked in perfect rhombohedral lattices, particle tracking simulations show that when the reagent is introduced into the column the beads would be displaced from the stack due to turbulent fluid flow and then resettle over the experimental time once the reagent flow stops. This allows enhanced (compared to the perfectly stacked scenario) and varying reagent access to many beads transiently as they settle down, which explains the spread of fluorescence intensities of beads reacted in columns in Fig. 4.

time ($\approx 300$ s). Even when the beads are floating up, there are many beads in proximity ($\approx 510$ beads in a region of size equivalent to a droplet in frame $t = 0.05$ s in Fig. 5) competing for the reagent molecules. The $\approx 100$ attomoles of initiators on the surface of the beads which are ideally packed ($t = 70$ s and $t = 300$ s in Fig. 4) in regular lattice arrangements (say in rhombohedral) have access to reagents in an interparticle void volume ($V_{void}$) of $4\left(\sqrt{2} - \frac{\pi}{3}\right) R_b^3 \approx 40$ fl[69,70]. The $5\,\mu$M concentration of fluorescently labeled nucleotides implies $\approx 0.2$ attomoles per void in a unit cell in a rhombohedral stacked lattice (Supplementary Fig. 11). On the other hand, in DBDR, the bead has access to the entire reagent volume within the droplet ($\approx 65$ pl) and $\approx 325$ attomoles of nucleotides within it. This is confirmed by the much faster diffusion time ($T_{diff, DBDR} \approx \frac{2R_d^2}{D} \approx 12.5$ s) of single nucleotides (diffusion coefficient $D \approx 10^{-6} - 10^{-5}$ cm$^2$/s[71,72]) across the droplet compared to the reaction time ($\approx 300$ s). On the other hand, diffusion across the length of the synthesis column ($h = 2.28$ mm, Supplementary Fig. 13) is expected to occur over much longer time scales ($T_{diff, col} \approx \frac{h^2}{2D} \approx 25000$ s). So, reagent molecules in the columns that are further away from the stack are less likely to contribute to the reaction by interacting with beads. Therefore, ideally the fluorescence intensity ratio ($FIR_{avg}$) between beads reacted using DBDR and in a perfectly stacked synthesis column is expected to be 500 (this can be the upper limit of attainable $FIR_{avg}$). However, due to the imperfect nature of the stacking and the non-zero contribution of diffusion our experiments yielded an $FIR_{avg} = 3.2$. Apart from this, the AC electric field driven migration of the charged reacting species in the microdroplet may also contribute to the improved reagent access to the solid supports in DBDR (Supplementary Note 6 and Supplementary Figs. 14–17). While our analysis does not eliminate the possibility of some beads reacted in synthesis columns having the same access to reagents as in DBDR, it emphasizes the inadequate access to reagents over a large number of beads that limits the overall fidelity of these macroscale reactors. Therefore, we see that stacking of beads in synthesis columns which is known to provide the large surface area of solid support for the synthesis of large quantity of oligonucleotides[1,2,4,5,8,11,12] also contributes significantly to reaction errors (reduced reaction fidelity). These errors accumulate over multiple synthesis cycles to ultimately negate the advantage of large synthesis quantity achieved in the columns. To compensate for this reduced reagent access and longer diffusion times in synthesis columns, higher reagent concentrations are used[1,8,11,12]. However, this increases reagent wastage per reaction that can accumulate over multiple synthesis cycles to make the synthesis of longer and more complex oligomer strands inviable. Many approaches have been employed on the macroscale, such as column agitation, to improve interfacing of solid supports with synthesis reagents for higher yields[73]. Though reliable improvements in reaction fidelity have been obtained in some cases, implementing them at scale would require considerable re-engineering of existing column and titer plate-based synthesizers. Consumption of large amounts of reagents also remains a challenge in these systems, especially for iterative design-build-test-learn approaches. On the other hand, by encapsulating one bead per droplet, DBDR ensures optimal reagent access to the surface of every bead during the experimental duration. Precise dielectrophoretic micromanipulation ensures that individual beads

have access to their own packet of reagents in their immediate vicinity thus eliminating the possibility of suboptimal reagent access to the beads (solid supports) as in synthesis columns. Although we demonstrated reaction in a single bead-droplet system, DBDR can open up the possibility of generating large-scale one-bead-one-compound[74] libraries while minimizing reagent usage for iterative experimental design cycles by combining parallel droplet generation[75,76] with parallel micromanipulation[77].

## Discussion

In summary, we have demonstrated that microbeads can be controllably encapsulated and then ejected from aqueous microdroplets to carry out high-fidelity solid-phase synthesis reactions. This process has many significances. Firstly, the demonstrated physical process of microbead encapsulation and ejection from microdroplets extends the capability of dielectrophoresis (and more generally any trapping and micromanipulation technique) to drive microparticle transport across immiscible fluidic interfaces, especially with sharp micron-sized radius of curvature. Secondly, this also extends the physical sample handling capability of droplet microfluidic platforms to include solid supports, thus opening a route for the droplet microfluidic implementation of solid-phase synthesis. The counteracting dielectrophoretic and capillary forces enable the voltage-controlled interfacing of individual microbeads (solid supports) with picolitre scale reagent microdroplets to form a bead-droplet reactor. As proof-of-concept, we demonstrated the enzymatic coupling of a fluorescently labeled nucleotide to the 3′ end of the initiator strand bound to the bead surface. Finally, fluorescence measurements suggest that reactions implemented using DBDR exhibit higher fidelities than synthesis columns due to the (i) absence of bead-bead stacking as in synthesis columns, and (ii) faster reagent diffusion across microdroplet reactors. We would also like to emphasize that DBDR is vastly different from previous reports of beads in droplet reactor systems in which the enzyme is immobilized on the bead and the reactants and products are confined to the droplet[57].

The proof-of-concept demonstration of enzymatic coupling of nucleotides onto the bead surface using DBDR with a 3.2-fold higher fidelity compared to synthesis column if repeated multiple times (say $N$) to generate $N$ nucleotides long oligomers across many beads. For such $N$ nucleotide long oligomers, the error free products ratio[14] (EFPR) synthesized using DBDR and synthesis columns is given by $EFPR = \frac{1}{10^6} \times 3.2^N$ (around a million beads are used in the synthesis column). Therefore, for merely $N = 12 > \frac{6}{\log_{10} 3.2}$, $EFPR > 1$ can be achieved. So, higher quantity of intended error-free synthesis product can be achieved on 1 bead using DBDR than on a million beads using synthesis columns for oligomers that are longer than merely 12 nucleotides ($N = 12$). Furthermore, the reagent usage ratio (RUR) between DBDR and synthesis columns would be $RUR = \frac{N \times 65\,pl}{N \times 50\,\mu l} \approx 10^{-6}$. Additionally, the synthesis time ratio (STR) between DBDR and synthesis columns would be $STR = \frac{N \times 12.5\,s}{N \times 25000\,s} = 5 \times 10^{-4}$. Therefore, DBDR can synthesize significantly higher quantity of error free strands while consuming much less reagents without the need for post synthetic purification or error correction at much faster rates. The synthesis fidelity improvement would be more pronounced with increase in $N$. This immense potential of DBDR for high fidelity solid-phase synthesis combined with reduced reagent consumption and faster reaction rate can be harnessed by combining parallel droplet generation[75,76], particle manipulation[77], and automation to scale up our proof-of-concept demonstration for high-throughput generation of solid-phase synthesis products. This can expand the accessible synthetic design space using solid-phase synthesis for a wide range of applications in chemistry, biology, medicine, material science and information technology. The high synthesis fidelity made possible by

DBDR is the foremost requirement for error free digital data storage in DNA[63,78]. Furthermore, instead of extracting the beads from the device, DBDR can be combined with existing approaches for the droplet microfluidic implementations of gene assembly, their transformation into cells, cell growths, and colony screening to develop an entirely droplet microfluidic-based design-build-test-learn cycle for synthetic biology[15] that could minimize reagent consumption in the otherwise resource-intensive iterative process.

DBDR can have applications beyond high-fidelity solid-phase synthesis with reduced reagent consumption and wastage. It can be a great way to exploit the basic advances made in microdroplet chemistry[79,80] for solid-phase synthesis on an integrated chip-scale droplet microfluidic platform for practical applications. It can also be an excellent platform to study enzymatic reactions in volume (within the microdroplet) and surface (on the bead surface) confined environments[81,82]. These possibilities make DBDR an exciting platform for on-chip chemistry in droplet microfluidic systems.

## Methods

### Device fabrication

Fabrication started with $\sim 500\,\mu m$ thick silicon wafers (p-type, $10 - 20\,\Omega\,cm$, $\langle 100 \rangle$) and borofloat glass wafers $\sim 550\,\mu m$ thick that are 100 mm in diameter.

The silicon wafer was etched in 5 layers to define the (1) alignment marks, (2) droplet generation microchannel, (3) reaction chamber (4) and (5) ports for external fluidic and electrical connections (Supplementary Fig. 1a–c). The first four layers were defined by reactive ion etching using $SF_6$ gas. In the 4th layer the ports were etched almost through the wafer from the backside. In the 5th layer, the ports were completed by laser drilling through the ports from the front side. The photoresist for each layer was patterned using a standard photolithographic approach with a maskless exposure system (Heidelberg MLA 150).

Eight hundred nanometer ($800\,nm$) of ITO was sputter deposited on piranha cleaned borofloat glass wafers at LGA Thin Films. This was followed by 2 layers of photolithographic processing: (1) the electrodes and alignment marks were defined by RIE of ITO from the entire wafer (barring the electrodes) using $CH_4$ and $H_2$ gases and (2) the contact pads were defined by evaporative deposition of 10 nm and 200 nm Gold followed by metal liftoff. After each layer, the wafer was wet cleaned using $5 : 1 : 1 :: H_2O : H_2O_2 : NH_4OH$ at $70\,^\circ C$ for 1 h (Supplementary Fig. 1d).

The glass and silicon wafers were then aligned (Supplementary Fig. 1e) and bonded anodically at $350\,^\circ C$ by applying a voltage of 350 V for 6 min. The wafers are then diced into 25 mm × 30 mm chips using a laser cutter (Supplementary Fig. 1f, g, and Fig. 1b). The chips were silanized using vapor phase deposition of Dimethyldichlorosilane (DDMS) at Integrated Surface Technologies to make the device interior surface hydrophobic for the formation of water droplets. The silane on the exterior surface of the chip was stripped using UV ozone treatment to stick fluidic connectors using Loctite 401 adhesives (Supplementary Fig. 1h). Electrical leads are soldered onto the gold contact pads through the electrical ports in the silicon wafer (Supplementary Fig. 1h).

### Experimental setup

The experimental setup consists of the device holder interfaced with the fluidic, electrical, and optical subsystems.

**Device sample holder.** A 3D printed plastic sample holder was used to mount the device on the experimental setup. Copper pins hold the device in place with a 0.17 mm glass coverslip (24 mm × 40 mm) from SPI below it. The sample holder was screwed onto a X-Y stage from Newport (Model#-406) mounted on a modified Nikon TE2000U inverted microscope (Supplementary Fig. 2a, b).

**Fluidic subsystem.** A piezo driven pressure controller (OB1 MK3+ from Elveflow) with a 30 psi input from the house nitrogen supply and a maximum output of 2000 mbar was connected to the input of a fluidic tank (15 ml plastic tube) using a 10 mm OD tubing (Supplementary Fig. 2c, d). The output of the tank flows into the device via a 1/16 inch OD and 1/32 inch ID polytetrafluoroethylene (PTFE) tubing from Masterflex (item#-EW06407-41) which was plugged into the fluidic port of the device using the Nanoport Assemby from IDEX Health & Science (part# - N-333).

**Electrical subsystem.** An A.C. high voltage amplifier (A. A. Lab Systems Ltd. Model#-A-303) amplifies the signal from a function generator (Hewlett Packard Model#-8116A) to generate a maximum output amplitude of 200 V (Supplementary Fig. 2e, f). The amplified output was connected to the electrical leads of the device through a single pole double throw (SPDT) switch which connects the supply across either of the electrodes ($E_1$ or $E_2$) and the ground pad.

**Optical subsystem.** The optical subsystem (Supplementary Fig. 2g, h) was built around a modified Nikon inverted TE2000U microscope. A blue LED (SOLIS-445C from Thorlabs, 445 nm and 5.4 W min) with a band pass excitation filter (D480/30x from Chroma) images the bead (6 μm diameter fluorescent green streptavidin coated polystyrene beads with excitation maxima at 441 nm and emission maxima at 486 nm, Catalog#-24157) and the fluid flow in the device onto a sCMOS camera from Thorlabs (Part#-CS2100M-USB). The experiments were recorded at 33 frames per second. A red LED (M625L4 from Thorlabs, 625 nm and 700 mW) excites the Alexa-647 to detect nucleotides labeled with the fluorophore (dCTP-AF647) in the reagent droplet. A sCMOS camera from PCO (PCO edge 5.5) was used to capture the low light intensity levels emanating from the nucleotides coupled to the initiator strands on the beads at 2 s integration time. Appropriate bandpass excitation (Item#-86-988 from Edmund Optics, 640 nm center wavelength, 14 nm bandwidth, OD - 6) and emission (Item#-86-987 from Edmund Optics, 676 nm center wavelength, 29 nm bandwidth, optical density - 6) filters were used to ensure non-overlap of the excitation and emission spectrum. A Nikon objective (ELWD-20, 20x mag, 0.45 NA) with a correction collar for spherical aberration correction (set at 0.7 mm which is 0.17 mm thick glass coverslip + 0.53 mm thick borofloat glass of the device) was used for imaging.

## Sample preparation

**Preparing the oil solution by adding surfactant.** 4 ml of Span 80 (S6760 from Sigma Aldrich) was added to 200 ml of 1 cSt silicone oil (PSF – 1 cSt from Clearco Products) to make a 2.5% $w/w$ solution. It was sonicated for 30 mins to ensure complete dissolution of the surfactant.

**Attaching initiator strand to beads.** The initiator strand, which was a biotinylated oligomer with 25 bases (T25mer, 5′ biotin, IDT) was attached to the 6 μm diameter ($R_b = 3$ μm) streptavidin coated green-fluorescent polystyrene beads using the strong biotin-streptavidin hydrogen bond. The reaction was carried out for 60 min at 2 °C, 14 RPM. The approximate starting yield (140 attomoles per bead (T25mer bound)) was determined by measuring the optical density at 260 nm (Nanodrop) before and after initial binding, then subtracting the supernatant and wash OD values from the starting yield. Binding/wash buffer: 20 mM Tris pH 7.5, 1 M NaCl, 1 mM EDTA, 0.0005% Triton-X 100 (45 μl (plus 5 μl 100 μM T25) for binding reaction, and 500 μl for wash steps). Based on a particle concentration of 1.4%, the number of beads was ~867,000 per 25 μl reaction (accounting for a 20% loss due to mixing and washing steps). Beads with initiator strands were then spun down using an Eppendorf Minispin (Catalog#-022620100) to remove the supernatant and were segregated into two parts (i) for Dielectrophoretic Bead-Droplet Reaction in the fabricated chip, and (ii) for benchtop synthesis in columns.

**Suspending initiated beads in oil solution.** The spun down initiated beads are suspended in the oil solution by sonication. The concentration of the beads in the silicone oil solution were tuned to ensure mostly a single bead floats in the vicinity of the electrodes within the field of view of the objective.

**Preparing reagent solution.** The reagent solution was prepared by mixing 25 μl of reagents consisting of the fluorescently labeled base (dCTP-AF647) in a buffer solution of 50 mM Potassium Acetate, 20 mM Tris-acetate, 10 mM Magnesium Acetate, and 0.25 mM Cobalt Chloride with the enzyme (TdT) solution consisting of 3 μl of 50 mM KPO4, 100 mM Sodium Chloride, 1.43 mM β-ME, 50% glycerol, and 0.1% Triton X-100 solution in an Eppendorf tube. This reagent solution was formulated by initial benchtop experiments as described in the subsequent experimental procedure section. A trace amount of sodium salt of fluorescein (F6377 from Sigma Aldrich) was added to the reagent using a toothpick to discriminate it from the continuous phase inside the microfluidic device.

**Filling device with oil solution as continuous phase.** The device was completely immersed in 50 ml of the 2.5% w/w solution of Span80 in 1 cSt silicone oil contained in a glass jar inside a vacuum desiccator. As the desiccator was evacuated the air inside the device was drawn out. When the desiccator is refilled with air, the silicone oil solution gushes into the device to fill it completely without any trapped air bubbles.

**Mounting device on sample stage.** The device was then removed from the glass jar, its outer surface was cleaned by thoroughly wiping with isopropanol, and then mounted on the sample holder (Supplementary Fig. 3a). The objective was focused on the output of the droplet generation channel and the ITO electrodes.

**Making electrical and fluidic connections to device ports.** Electrical connections are made from the output of the amplifier to the ground pad and to the trap electrodes through the SPDT switch (Supplementary Fig. 3a). The fluidic tank was filled with 15 ml of the above oil solution. Pressure was applied using the pressure controller to fill the output PTFE tubing from the tanker with oil solution which is dipped at the other end inside the 1.5 ml tube containing the reagent. Just before oil starts dripping from the tubing into the reagent tube, the height of the PTFE tubing and reagent tube were raised to suck the reagent into the tubing. Then the tubing was lowered again into another Eppendorf tube containing the oil solution. As the reagent solution started dripping, the height was raised again to fill the PTFE tubing with the oil solution while ensuring there are no trapped air bubbles. The tubing was then connected to the device inlet while pushing out the oil solution at the bottom to ensure no air gaps and fluidic continuity (the oil solution inside the device and at the bottom of the tubing are the same). This approach prevented immediate flow of the reagent solution through the droplet generation channel as soon as the PTFE tubing was connected thus allowing time for experimental setup and control (Supplementary Fig. 3b).

## Experimental procedure for bead-droplet interaction

**Encapsulation and ejection of bead from droplet.** Beads suspended in the oil solution were introduced into the device through the oil inlet (Supplementary Fig. 1a). The voltage supply was switched on and set to around 40 V amplitude at 200 Hz to dielectrophoretically trap a bead floating near the top electrode. Then the voltage supply was switched off. Following this a pressure of 60 mbar was applied on the pressure controller to drive reagent flow in the device. As the reagent approached the entrance of the microfluidic channel, a sudden pressure pulse of approximately 10 mbar for approximately 300 ms was exerted to dispense a single droplet into the reaction chamber (Supplementary Movie 5). A higher-pressure pulse or a longer pulse duration would

lead to the generation of multiple droplets (Supplementary Movie 6). Then the voltage supply was switched on again at 20 V amplitude and 200 Hz to trap the droplet adjacent to the bead on the top electrode.

At this point the supply voltage was gradually increased to ~ 120 V amplitude. The droplet moves toward the electrode to encapsulate the bead. Subsequently the voltage was reduced to 0.1 V amplitude and the bead was ejected out of the droplet. Supplementary Movie 1 depicts the encapsulation and ejection process. The bead can undergo multiple encapsulations and ejection (Supplementary Movie 7) as well by switching the voltage between high and low. The images extracted from these videos were suitably processes for enhanced clarity without altering the data and results.

**Estimating interfacial tension and contact angle.** The interfacial tension values reported in the main text were measured using the standard Wilhelmy plate method. The contact angles were measured by capturing the droplet shape on a streptavidin coated glass slide (GS-SV-5 from Nanocs) and silanized glass surface and then estimating the angle it forms on the surface through shape fitting.

**Estimating electrical conductivity of reagent.** The conductivity of the reagent droplet was measured using an Orion 3 Star Conductivity Portable.

## Modeling of bead-droplet Interaction

**Coupled electrohydrodynamic simulations.** The electric field driven encapsulation of the microbead into the microdroplet and its ejection out of it was modeled by using coupled electrohydrodynamic simulations in COMSOL Multiphysics (Supplementary Fig. 4). The fluid flow (aqueous droplet motion in silicone oil medium) was modeled as two-phase fluid flow using the Navier-Stokes equation[45]. The phase variable $\phi$ tracked the fluidic interface between the droplet and the suspension medium using the phase field method[83]. $\phi = -1$ within the silicone oil medium and $\phi = 1$ in the droplet. It transitions from $-1$ to $1$ at the interface of the droplet and the suspension medium. To model the driving electric force, the electric charge continuity equation[45,51,84] was used to solve for the non-uniform electric field distribution when an AC voltage was supplied across a pair of electrodes of different dimensions. The electric force density on the fluids was evaluated by using $\vec{\nabla} \cdot \overleftrightarrow{T}$. Here, $\overleftrightarrow{T}$ is the Maxwell Stress Tensor[44,51,84–87]. This force sets the fluid flow in motion. As the droplet moves, the boundary condition of the electric charge continuity equation changes. This changes the electric field distribution and $\vec{\nabla} \cdot \overleftrightarrow{T}$ in turn. This underlines the basic coupling between the Navier Stokes equation and the charge continuity equation. In the simulations, the bead is modeled as a stationary polarizable dielectric particle with a fixed contact angle ($\theta = 145°$) that the droplet forms on its surface in the silicone oil medium. This leads to an additional capillary force acting on the bead-droplet system when the bead and the droplet are in contact. For simplicity, axis symmetric simulations were adopted. This helps focus on the essential physical interaction without getting into the nuances of device design.

The encapsulation and ejection process can be modeled as a balance between the electric and capillary force. To understand the scaling laws underlying these forces, we derived approximate analytical equations for the electric and capillary forces.

**Dielectrophoretic force.** The dielectrophoretic force[44,51,84–87] on a generic charge neutral polarizable particle (of radius $R_p$) can be evaluated by expressing its polarizability and the non-uniform electric field as an infinite series of multipolar expansions[87]. We base our approximate analysis on the dipolar/first ($n = 1$) term of this series. The approximate dielectrophoretic force ($\vec{F}_{DEP}$) then scales as $\vec{F}_{DEP} \propto K_1 R_p^3$. Here $K_1$ is the familiar Classius-Mossotti factor and $R_p$ is the radius of the polarizable particle which can be the droplet or the bead. $K_1$ is given by $\frac{\tilde{\varepsilon}_p - \tilde{\varepsilon}_o}{\tilde{\varepsilon}_p + 2\tilde{\varepsilon}_o}$. Here $\tilde{\varepsilon}_{p/o}$ is the complex permittivity of the particle or the suspension oil medium which is a function of their respective relative permittivities ($\varepsilon_{p/o}$) and conductivities ($\sigma_{p/o}$). Therefore, the water droplet experiences a much larger force than the bead due to its larger permittivity contrast with the oil medium as well as much larger size (Supplementary Fig. 5). So, if multiple droplets are suspended in the reaction chamber, the primary electric field driven effect is the merger of the droplets.

**Capillary force.** The capillary[45,51,52] force on the bead-droplet system arises due to the change in the total interfacial energy of the system as the bead is encapsulated/ejected within the droplet. For our case in which $R_b \ll R_d$, the change in total interfacial energy is given as $\triangle U_{IT} \approx - \gamma_{ow} \cos\theta \triangle A_{ws}$. Here, $\gamma_{ow}$ is the oil-water interfacial tension and $A_{ws}$ is the interfacial area of the water/aqueous droplet and the solid bead surface (Supplementary Fig. 6). From this the scaling of the interfacial capillary force can be approximated as $\vec{F}_{IT} \propto R_b \gamma_{ow} \cos\theta$.

## Experimental procedure for chemical coupling
**Chemical coupling of base and control on the device.** The above physical process was used (with a 300 s encapsulation time) for the enzymatic coupling of fluorescently tagged nucleotides onto the initiator strand on the bead with a few additional intermediate images of the area around the reaction zone captured as enlisted below.

- Before loading the beads into the device an image to estimate the background noise under red illumination (Supplementary Fig. 7a, b).
- After the bead was dielectrophoretically trapped on the top electrode an image each with the blue and red excitation are captured to measure the level of the red fluorescence signal from the site of the bead just prior to the reaction (Supplementary Fig. 7c, d).
- Finally, after the encapsulation and ejection process another set of images under blue and red excitation were captured (Supplementary Fig. 7e, f).

These images were captured as 16-bit Tiff files. Supplementary Fig. 7c, f are used in Fig. 4a of the main text. These steps were repeated to see the repeatability of the chemical coupling reaction on our platform. The three different reacted beads under red excitation are depicted separately in Supplementary Fig. 7g and are also used in Fig. 4 of the main text.

The control experiment was repeated using the above process but with beads without initiator strands and reagent droplets without the enzymes (Supplementary Fig. 7h–m). Supplementary Fig. 7h, m are used in Fig. 4b of the main text.

**Column synthesis.** Firstly, free solution reactions are implemented to develop optimal room temperature protocol (Supplementary Fig. 8a, b) for translation into DBDR. Results were analyzed using reverse-phase high performance liquid chromatography (HPLC) (Supplementary Fig. 8b).

To simulate an enzymatic synthesis reaction using a column, an open-top nylon syringe filter (Omicron SFNY04XB, 4 mm, 0.45 µm) was used. To the bottom filter (0.45 µm pore size) which was held in place by a plastic ring, 15 µl of beads (1.2 M) were added. A top filter was then positioned above the reagent bed. Between the filters the reaction volume was about 15 µl. A 1 ml syringe was used to push the bead medium (10 mM HCL, 2 M NaCl, 1 mM EDTA, 0.0005% Triton-X 100, pH 7.3) passed the bottom filter until it completely exited the drip director. 50 µl (6 µl TdT (20 U/µl), 5 µl 10$x$ TdT buffer (50 mM potassium acetate, 50 mM Tris-acetate, 10 mM magnesium acetate, pH 7.9 @ 25 °C), 5 µl 10× (2.5 mM) solution of CoCl$_2$, 0.25 µl 1 mM Alexa Fluor™ 647-aha-dCTP, 33.75 µl water) were added to the top filter, and an empty 1 ml syringe was used to push the reagent passed the top filter (flow rate at 50 µl per second), into the reaction area until the

reagent could be seen inside the drip director. The column was kept upright for 5 mins. Once the reaction was completed, the empty 1 ml syringe was used to push the spent reagents through the column until the drip director was clear. Afterwards, 1 ml of bead suspension medium (described above) was used to wash the beads. These beads were then taken in a 1.5 ml Eppendorf tube for analysis using fluoroscopy.

As a control, synthesis was performed on beads without initiators and reagents without TdT and the reaction was analyzed through fluorescence measurements.

**Measuring fluorescence from beads reacted in columns.** About 3 μl of the reacted bead suspension in the buffer was taken in an Eppendorf tube and was diluted to ensure the bead concentration was small enough to prevent signal interference from beads in different planes while being large enough to have ample beads within the field of view to get a statistically significant inference about fluorescence intensity distribution. The beads were introduced into the device filled with MilliQ water. The same chips were used for fluorescence measurements to ensure identical optical environment for comparison between on-chip experiments with their column counterparts. Once the beads settled down (imaged using blue excitation) the excitation was switched to red to image the fluorescence intensity of the beads. Many such frames of red fluorescent beads were collected with a large number of beads (285). Some are shown in Supplementary Fig. 9a. A few representative beads spanning the entire range of fluorescence intensities are used in Fig. 4d of the main text.

The control experiments implemented using DBDR (Fig. 4b of main text) were reiterated on the columns. The fluorescence of these beads was measured following the same procedure as discussed in the previous paragraph (Supplementary Fig. 9b).

### Image processing and data analysis
**Encapsulation and ejection of bead from droplet.** The recorded video (Supplementary Movie 1) of the encapsulation and ejection process was analyzed frame by frame using ImageJ and snapshots that best represent the processes were selected and labeled for Fig. 2 of the main text.

**Establishing enzymatic coupling of base to the initiator strands on the bead in DBDR.** Maintaining the same scale of $250 - 3300$ across the red fluorescence images, the difference in brightness of the bead with enzymatic coupling (Fig. 4a main text and Supplementary Fig. 7f) and the control bead (Fig. 4b main text and Supplementary Fig. 7m) was obvious.

**Analyzing fluorescence intensity distribution.** Each frame in Supplementary Fig. 9a was analyzed using predefined image processing functions in Matlab to detect the beads (Supplementary Fig. 10), binarize them, evaluate their mean fluorescence intensity, and evaluate fluorescence intensity distribution across a horizontal line passing through the bead center. Average fluorescence values across frames were collected to plot a histogram of the fluorescence intensity distribution of all beads reacted on the benchtop using synthesis columns.

The average fluorescence intensity of the three beads reacted using DBDR (Supplementary Fig. 7g) were found to be 2184.5, 2171.7, 1816.2 (same unit as was used for column fluorescence data). These values are higher than the fluorescence intensity (represented in Supplementary Fig. 10d) of all the 285 beads reacted in columns whose fluorescence data was collected. This indicates that the reaction fidelity using DBDR is higher than synthesis columns.

**Analyzing statistical distribution of data.** To establish the statistical significance of our fluorescence comparison-based claim that the solid-phase synthesis reaction fidelity achieved using DBDR is higher

than synthesis columns we resort to statistical hypothesis testing. We seek to establish that the mean fluorescence intensity of beads reacted using DBDR is higher than the mean fluorescence intensity of beads reacted using synthesis columns at significance level of 0.05 or a confidence level of 95%. The t-test which tests for the null hypothesis of equivalence of sample means for both the synthesis methods using the following test statistic[88–92] (t value) is appropriate for our purpose.

$$t = \frac{\mu_{DBDR} - \mu_{Column}}{\sqrt{\frac{\sigma_{DBDR}^2}{N_{DBDR}} + \frac{\sigma_{Column}^2}{N_{Column}}}} \tag{1}$$

Here, $\mu_{DBDR/Column}$ is the sample mean of the respective synthesis methods, $\sigma_{DBDR/Column}$ is the sample standard deviation of the respective synthesis method, and $N_{DBDR/Column}$ is the number of bead samples over which the mean and the standard deviation were evaluated in the respective synthesis methods. For columns, $N_{column} = 285$. This is the total number of beads that were accounted for in the histogram in Supplementary Fig. 10d. For DBDR, $N_{DBDR} = 3$. These are the 3 beads represented in Supplementary Fig. 7g. The relevant values are summarized in Supplementary Table 2a. As the standard deviations and the number of bead samples are unequal in the two synthesis methods, we use the Welch's t-test for the statistical significance analysis[89,90,93]. The degree of freedom for the Welch t-test which is given by the Welch-Satterthwaite equation is[90,94,95]:

$$df = \frac{\left(\frac{\sigma_{DBDR}^2}{N_{DBDR}} + \frac{\sigma_{Column}^2}{N_{Column}}\right)^2}{\frac{\left(\frac{\sigma_{DBDR}^2}{N_{DBDR}}\right)^2}{N_{DBDR}-1} + \frac{\left(\frac{\sigma_{Column}^2}{N_{Column}}\right)^2}{N_{Column}-1}} \tag{2}$$

The values of $t$ and $df$ evaluated using Eq. 1 and eq. 2 are 11.65 and 2.14 (summarized in Supplementary Table 2b). Therefore, $2 < df = 2.14 < 3$. Using a standard t-test table for two-tailed testing we see that if $df = 2$ for a two-tailed significance level ($\alpha$) of 0.01 the critical t-value ($t^*$) is $9.925 < t$ and for a two-tailed significance level ($\alpha$) of 0.002 the critical t-value ($t^*$) is $22.327 > t$. On the other hand, if $df = 3$ for a two-tailed significance level ($\alpha$) of 0.002 the critical t-value ($t^*$) is $10.215 < t$ and for a two-tailed significance level ($\alpha$) of 0.001 the critical t-value ($t^*$) is $12.924 > t$. Thus, we can safely say that our null hypothesis can be rejected at significant level of $\alpha = 0.01$ or at a confidence level of 99%. Hence, our result is definitely significant at $\alpha = 0.05$ or a confidence level of 95%. This was confirmed using the inbuilt ttest2 function in Matlab for Welch's t-test which rejected the null hypothesis. To evaluate if the sample sizes ($N_{DBDR}$ and $N_{column}$) were sufficient for statistical testing, we calculate the power of the statistical test[96] using the inbuilt sampsizepwr function in Matlab for $\alpha = 0.05$. We obtain a statistical power of almost 1 (a power of 0.8 at $\alpha = 0.05$ is generally considered adequate[96]). Therefore, our sample size suffices for statistical testing.

The Welch's t-test, which is a parametric test is generally robust for normal distributions with unequal sample sizes and standard deviations[89,90,93]. For deviations from normal distributions (Supplementary Fig. 10d), nonparametric tests (which do not assume any specific distribution profile) operating on the ranks of the experimentally observed values rather than the actual values themselves are more robust[93,97]. It is established in statistical literature that a rank transformation on the conventional Welch's t-test would counter the combined effects of unequal standard deviations as well as non-normal distributions[93]. Therefore, we apply the above statistical testing procedure to the combined ranks of the average fluorescence intensities of beads reacted using the synthesis column and DBDR. The 3 beads reacted using DBDR have higher fluorescence intensities than the 285 beads reacted in synthesis columns. So, the beads reacted in columns have ranks from 1 to 285. While the beads reacted using DBDR have ranks from 286 to 288. The respective means ($\mu_{DBDR/column}$) and

standard deviations ($\sigma_{DBDR/column}$) are summarized in Supplementary Table 3a. Using Eqs. 1 and 2, we evaluate $t = 29.41$ and $df = 285.79$. We see that at $\alpha = 0.05$, $t^* = 1.9683 < t = 29.41$ for $df = 285$ or 286. Therefore, the null hypothesis of equivalence of means of DBDR and columns can be safely rejected at the confidence level of 95%. The inbuilt ttest2 function in matlab confirms this. Furthermore, a statistical power of 0.9729 is obtained which confirms that the sample size suffices for the statistical inference. The results are summarized in Supplementary Table 3b.

**Modeling of bead stacking in column and comparison of reagent access with bead-droplet reactor**
**Analytical modeling of ideal case of beads stacked in perfect lattices.** To understand how stacking of beads in synthesis columns limits reagent access to the bead surfaces, we begin by considering the ideal case scenario of stacking of beads into perfect lattices. In such a case, effectively one bead is enclosed in a unit cell and access the reagents within the void of the unit cell. The simple cubic is one of the most loosely packed lattices while the rhombohedral is one of the most tightly packed[67]. The volume of a bead of radius $R_b$ is $\frac{4}{3}\pi R_b^3$ (Supplementary Fig. 11). The volume of a simple cubic unit cell is $8R_b^3$ while that of a rhombohedral unit cell is $4\sqrt{2}R_b^3$. Therefore, the void volume in these unit cells is $4\left(2 - \frac{\pi}{3}\right)R_b^3 \approx 103\,\text{fl}$ and $4\left(\sqrt{2} - \frac{\pi}{3}\right)R_b^3 \approx 40\,\text{fl}$ respectively[69,70]. The reagent solution has 5 μM concnetration of nucleotides ($2.5 \times 10^{-7}$ mmoles of nucleotides were added to 50 μl of reagent solution). Therefore, within the void volume of the simple cubic and rhombohedral unit cells there would be around $103\,\text{fl} \times 5\,\mu\text{M} = 0.52\,\text{attomoles}$, and $40\,\text{fl} \times 5\,\mu\text{M} = 0.2\,\text{attomoles}$ of fluorescently labeled nucleotides respectively which is $\approx 280$ and $\approx 700$ times more than the number of nucleotides in the voids of the simple cubic and rhombohedral lattice respectively. On the other hand, the volume of the droplet ($R_d = 25\mu m$) which forms a contact angle ($\beta$) of $138°$ on the device surface is $\frac{(2 - 3\cos\beta + \cos^3\beta)}{3}\pi R_d^3 = 65\,\text{pl}$[51,52]. At the same 5 μM concentration of fluorescently labeled nucleotides, the nearly 100 attomoles of initiator strands will have access to $65\,\text{pl} \times 5\,\mu\text{M} = 325\,\text{attomoles}$ of fluorescently labeled nucleotides within the droplet.

**Particle tracking modeling for imperfect tracks.** For simplicity of modeling as well as to reduce memory and time requirements, a 2D mirror symmetric simulation was set up combining turbulent fluid flow and particle tracking simulations. The simulation space mimicked the dimensions of the reaction column. A rhombohedral stack of beads was defined at the bottom of the simulation space. Fluid flow equations for turbulent flow[68] were simulated to mimic fluid flow in the column when reagents are introduced into it. This fluid flow exerted a drag force on the particles which set them in motion within the column dispersing the perfect stack in the process. The beads have increased and varying access to reagents. Once the reagent inflow subsides, the beads eventually settle down under the influence of gravity into imperfect stacks. Further modeling details can be found in Supplementary Note 5 and Supplementary Fig. 12.

**Reagent diffusion.** The synthesis columns used in the reaction are 2.28 mm long whereas the microdroplets used in DBDR are 50 μ$m$ ($5 \times 10^{-5}$m) in diameter (Supplementary Fig. 13). The diffusion time is approximated as $t_D \approx \frac{l^2}{2D}$[45]. Here, $D$ is the well-known diffusion coefficient which is $\approx 1.3 \times 10^{-5}$ cm$^2$/s[71,72] for single nucleotides. Using this we obtain $t_{D,col} \approx 2000\,\text{s}$ and $t_{D,DBDR} \approx 1\,\text{s}$.

**Electric field driven enhancement of reagent concentration in DBDR.** The reagent droplet in DBDR consists of many positive and negatively charged species which are necessary for the enzymatic coupling of nucleotides to the initiator strands. The applied AC electric field for dielectrophoretic trapping of the droplet and bead also drives ion migration closer to and further away from the encapsulated bead during alternate phases of the AC cycle. This changes the time averaged concentration of ions that the initiators on the bead are exposed to over an entire AC cycle. This migration of ions is modeled using the Nernst-Planck equation[45,98] in COMSOL Multiphysics. We model the effect of the AC supply voltage amplitude and frequency. Details can be found in Supplementary Note 6, Supplementary Table 4, and Supplementary Figs. 14–17.

## Data availability
All data generated and/or analyzed in this study are included in the manuscript. Source Data is available as a Source Data file. Source data are provided with this paper.

## Materials availability
Material requests should be addressed to P.P. or M.A.J.

## Code availability
MATLAB codes used to analyze data have been provided as .m files with supporting Readme files together with the Source Data file.

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

## Acknowledgements

We thank Dr. Paul Christopher Hansen for helpful discussions on numerical simulations. This project was funded by National Human Genome Research Institute (NHGRI) and National Institute of General Medical Sciences (NIGMS) of the National Institute of Health (NIH) under grant numbers 5R21HG009758 and R01GM138716 respectively and by the Stanford SystemX Alliance Seed Funding. Part of this work was performed at the Stanford Nano Shared Facilities (SNSF)/Stanford Nanofabrication Facility (SNF), supported by the National Science Foundation under award ECCS-2026822.

## Author contributions

L.H., R.W.D., P.P., M.A.Z., and M.A.J. conceived the project. L.H. supervised the project. P.P. designed and fabricated the device with process inputs from M.A.Z., Y.C., and L.G. P.P. and M.A.Z designed and built the fluidic, electrical, and optical setups. P.P. designed and conducted the on-chip experiments and data analysis. M.A.J. performed chemical sample preparation and column synthesis experiments. P.P. performed the numerical simulations. M.A.Z. wrote the image processing codes to analyze bead fluorescence. M.W. helped with experimental data analysis. Y.H. performed the contact angle and interfacial tension measurements. P.P. wrote the manuscript with significant inputs from M.A.Z., M.A.J., and L.H.

## Competing interests

The authors declare no competing interests.
