## [Peer Review File · Nature Communications]

Dielectrophoretic bead-droplet reactor for solid-phase synthesisREVIEWER COMMENTS

Reviewer #1 (Remarks to the Author):

This manuscript describes a promising method based on dielectrophoretic bead-droplet interaction to propose enzymatic nucleotide synthesis. The authors demonstrate the principle of the beads encapsulating and ejecting process in detail and show its superiority over conventional synthesis columns. The reaction performance is markedly enhanced in this practical manner. Here, I humbly recommend a "major revision" in light of some comments to refine the manuscript.

1. The authors mention that DBDR achieves near-perfect fidelity with a $FIR_{avg}=3.2$ better than columns in practice, then calculating the columns' perfect stacking situation seems less critical. It's unlikely to occur and has no relation to DBDR. Please consider abbreviating it or present in the supplementary file.
2. The interaction between droplets and beads is the core of this study. The readers will likely be interested in its design principle and the impact of changes in critical parameters (Fig S4). Please consider strengthening the discussion of these in the manuscript.
3. Fig 3 shows the change in the fluorescence signal compared to the control group, and its reaction principle has already been mentioned in Fig 1. Please consider merging Fig 3 with Fig 4.
4. Can multiple rounds of nucleotide synthesis be achieved using the current DBDR system? Also, how to recover the bead with the nucleotide sequence from the many unreacted polyethylene particles intelligently? It seems to be easily lost.
5. What will happen if two or more beads are in the reaction area facing a droplet? It should be a frequently encountered situation in experiments.

Reviewer #2 (Remarks to the Author):

The authors describe a simple and elegant solution to the long-standing challenge of extending solid-phase synthesis to droplet-based microfluidics, using di-electrophoresis to reversibly introduce beads into reagent droplets. By adjusting the bias on a trap electrode, the balance between dielectrophoretic and capillary forces is modified, enabling controlled transfer of the beads. Experimental control is clearly demonstrated, and observed results are broadly consistent with electrohydrodynamic simulations. The technique is successfully applied to enzymatic coupling of single nucleotides onto initiator strands bound to a microbead, with the authors showing improved fidelities relative to an equivalent column-based experiment. Overall, the paper is well written, with extensive supporting data. It is an innovative and useful addition to the toolkit of techniques for droplet microfluidics, and I am happy to recommend it for publication.

Reviewer #3 (Remarks to the Author):

I really struggled to understand what the authors actually did, and to find the evidence for this. I think that they made a microfluidic droplet containing red fluorescent labelled reactant solution, and manipulated a bead with immobilized green fluorescent labelled substrate into the droplet using an electric field, and then pulled the bead out after 300s to show a high level of this single coupling reaction. There were a lot of diagrams and long overly wordy written sections, but very little data to support this, and I really struggled to understand what was actually done.

This needs a major condensation re-write, and the removal of a lot of the advantage claims of this techniques over existing technologies, as they are not relevant to what is demonstrated in this work. The authors have done something really cool and interesting, they don't have to go over the top with selling it as the manuscript does in its current state. It is far too long. A lot of the information is repeated, especially on the overselling.

There are too many references – can these be curated to be more selective towards targeted relevant or seminal publications.

Introduction

- Too long – a 3 page introduction. Cut it down by at least half.
- A part from the increase in the need for more reagents and solvent waste, the disadvantages listed for column solid-phase synthesis are also limitations of microfluidic-cum-electrophoretic device manipulation synthesis (e.g. labor and time intensive sample handling, complexity of automated systems – your method is far from automated, facile or speedy) – be true about the advantages of what you have done.
- Your method has disadvantages – you only couple 1 thing! You cannot sell that it can be used to couple long oligomers as many times as you do. You can mention that it could be developed to allow this in the future, and the high success rate of the coupling reaction will facilitate this, but at the moment this is not something that should be repeatedly mentioned throughout the manuscript – it is massively overpromising something that has not been demonstrated.
- Other disadvantages are that at no point do you recover the coupled material – so this can only be used for applications where the materials can remain attached to the bead – do you even recover the beads? This was not clear.
- It is not high throughput or parallelizable in this form, here the proof of concept is applicable – could this be developed for a parallelizable coupling or multistep process – move to conclusions and outlook from the intro.

Device and Reaction Design

- This contains little information for the length. Again it is too long.
- How were the flow conditions optimized? What were they optimized to? Did this vary between devices? Operating range / values of flow rates, diagram of nozzle dimensions, chip height – is the chip 15 μm high, but the droplets have a 25 μm diameter, so the droplets are not spheres but are disks – was this allowed for in the volume calculations?
- Supp F1 – actual picture of the diagram next to the diagram.

Physical Working Principles of DBDR

- Too long (2 ½ pages). Condense this down into salient information (1-2 paragraphs) and expand the supplementary information on this. I am not a mathematician so am not able to comment on these aspects, but it seems that a lot of the information in this section is repeated in the supplementary information. Please still include it, but summarize it in the main paper, and expand the supplement for the interested reader.

Enzymatic Coupling of Nucleotide in DBDR

- How did you choose this optimal droplet size? You give an overly long explanation of this without showing the evidence of “failed” experiments that you used to reach the optimized conditions – these optimizations should be included in the supplementary information.
- “triplicated” – do you mean that you injected, reacted and retrieved a bead 3 times in 3 separate droplets and then averaged these fluorescent image results? Or did you do it loads of times and pick the best 3? In either case, this is not enough times to jump to the conclusions you have from here. If it is more, show the data and how you calculated the values and error in the values.
- Explain how challenging this technique is by showing what failure looks like, and being honest about how much skill and effort it takes to do this properly, and how you tell if it worked or not. For example, in AFM protein folding and unfolding there are thousands of events recorded, of which only a small percentage show folding / unfolding traces. The only reason this technique has value is because those that developed it were honest about the failure rate and how to do statistically relevant repeats to get the information from the technique. I cannot gauge this information from this manuscript, and this makes me skeptical.
- You don't recover your coupled molecules, so the low error rate on coupling doesn't really matter in comparison to other nucleotide coupling techniques – this issue crops up repeatedly, and is mentioned at the top.

Coupling Fidelity Enhancement in DBDR

- My main issue with this section is that there are many many ways of optimizing the control process to improve the yield. For example, mixing the beads on a cheap blood wheel mixer, as is commonly used in manual solid-phase peptide synthesis, or sparging the reagents so the column is only packed upon recovery, would significantly improve mixing, access of reagents to reactant surface, and thus the fidelity of the reaction. This control is deliberately making the worst possible result using an existing technology to make the DBDR process look better. It is way too long, and because the work doesn't recover the material, they are unable to compare the coupling fidelity to published experimental techniques.

- For example, later they mention that microarrays are not as good at this process. LeProust et al. (2010) *Nucleic Acids Research*, Volume 38, Issue 8, Pages 2522–2540 <https://doi.org/10.1093/nar/gkq163> synthesize 150mer oligos on arrays with a yield of 1 pMol full-length product with 4-5% yield from the crude synthesis by adjusting the reaction conditions to minimize detrimental side reactions. I cannot make comparisons between these works, because the current authors (1) have only demonstrated one coupling reaction and (2) have not quantified the yield in any way as they have not recovered the material. This is an example of overclaiming that needs to either be better quantified or removed completely. You have not made long oligos, so you need to dial down the number of times you say that this technique can make long oligos, because at the moment it can't.
- Discussion of electric field surface charge enhancement of reactants at the surface is difficult to follow and seems selective and hand-wavy rather than a complete explanation.

Implications and Outlook

- Move the long oligo stuff. You haven't done this and a whole paragraph on it is too much.
- Start with conclusions about what you have actually done, and then do the quick calculation of how this could be applied if the developments were made to allow multiple coupling reactions onto the same bead and recovery of the materials.

Conclusion

- "Near perfect" you haven't quantified this in any comparable way, and you haven't collected the material to measure this.
- "Fluorescence measurements suggest..." they do, lead with this, this is what you have measured.
- "We have achieved the first demonstration..." Liu et al. (2021) *Biomicrofluidics*, Volume 13, Issue 3, 034103, DOI: 10.1063/5.0050440 use pico injection to inject beads, react them and retrieve them. You are the first to do this with an applied field though, so say that.
- Vague mention of "...applications beyond solid-phase synthesis." What are these?
- Again, dial down the big claims and sell what you have actually done honestly.

Figures

1. These are all schematics, except (c) – I had to search really hard for this – move (a) and (d) to supplementary, make (c) bigger, are the arrows a calculation? Of what (the electric field)? This is not clear.
2. Are the red and blue pictures in (e) from image analysis or calculation? There is no scale on the images, is this the results of the beads not going in droplets that are too big/small? Refer to this in the text, I couldn't find this either. Put the applied field graph above the images so that you can see the image that corresponds to the applied voltage more easily.
3. Where is the graph of the distribution that is shown for the control in F4? This is just unscaled pictures and a diagram showing what you think it happening.
4. As mentioned above, there are issues with these experiments.
5. I don't really know what F5 is adding, again there are a lot of diagrams and not a lot of data in here.

Contents

	Page
A. Reviewer-1	1-6
1. Question-1	1
2. Question-2	2
3. Question-3	3
4. Question-4	4
5. Question-5	5
B. Reviewer-2	6-7
C. Reviewer-3	7-31
I. Introduction	9-12
1. Question-1	9
2. Question-2	9
3. Question-3	10
4. Question-4	11
5. Question-5	12
II. Device and Reaction Design	12-15
1. Question-1	12
2. Question-2	13
3. Question-3	14
III. Physical Working Principle of DBDR	15-16
1. Question-1	15
IV. Enzymatic Coupling of Nucleotide in DBDR	16-22
1. Question-1	16
2. Question-2	17
3. Question-3	18
4. Question-4	21
V. Coupling Fidelity Enhancement in DBDR	22-24
1. Question-1	22
2. Question-2	23
3. Question-3	24
VI. Implications and Outlook	24-25
1. Question-1	24
2. Question-2	25
VII. Conclusions	25-28
1. Question-1	28
2. Question-2	26
3. Question-3	26
4. Question-4	27
5. Question-5	28

VIII.	Figures	28-31
	1. Question-1	28
	2. Question-2	29
	3. Question-3	29
	4. Question-4	30
	5. Question-5	30

Reviewer-1

We are thankful to the reviewer for seeing the merit in our 1) fundamental physical demonstration of the encapsulation and ejection of a bead from a droplet, 2) its utility in solid-phase phase synthesis as evidenced by the demonstration of enzymatic coupling of single nucleotides onto individual beads functionalized with suitable initiator strands and 3) the improved fidelity of coupling reactions implemented using our proposed method over traditional synthesis columns. Furthermore, we appreciate the reviewer for his/her valuable input that we believe will help strengthen the manuscript and our research results. Below we address the specific concerns raised by the reviewer.

Question 1

The authors mention that DBDR achieves near-perfect fidelity with a $FIR_{avg} = 3.2$ better than columns in practice, then calculating the columns' perfect stacking situation seems less critical. It's unlikely to occur and has no relation to DBDR. Please consider abbreviating it or presenting it in the supplementary file.

Our Response

Explanation

The reviewer has made a very valid comment here. The $FIR_{avg} = 3.2$ compares reactions implemented in DBDR with practical columns in which beads would be stacked in improper lattice arrangements. There is no direct experimentally observed relationship between coupling reactions in the perfectly stacked configuration of beads in synthesis columns and DBDR. However, we included the perfectly stacked configuration to explain the observed $FIR_{avg} = 3.2$ based on well-established physical principles of reagent access to beads in stacked configurations. Existing literature, as is cited in the manuscript (references 69 and 70), discusses this in terms of regular lattice arrangement of beads. So, we used that as a starting point for analytical comparisons of reagent access to bead surfaces. Then we showed using particle tracking simulations in turbulent flow that the regular lattice arrangement (perfect stacking) of beads is an ideal impractical scenario. Our models show that even if the particles were to be arranged in regular lattice configurations to begin with, as reagents are introduced into the columns, the beads would rearrange into irregular lattices whereby the reagent access to each bead would be different.

Action

The following actions have been taken to address the reviewer's concerns:

- We have significantly reduced the discussion on ideal stacking in the **second paragraph of the coupling fidelity enhancement in DBDR** section of the main text. There we have primarily focused on comparison of practical improper stacks with DBDR. In the main text, a summary of the perfect stacking scenario is included to maintain analytical continuity without directly jumping to the complex case of randomly stacked beads due to turbulent flow of reagents in a synthesis column.

- The details of the ideal stacking have been moved to **section 6 of supplementary information** titled, “Comparison of reagent access to the bead surfaces in stacked bead configuration versus in DBDR”.

Question 2

The interaction between droplets and beads is the core of this study. The readers will likely be interested in its design principle and the impact of changes in critical parameters (Fig S4). Please consider strengthening the discussion of these in the manuscript.

Our Response

Explanation

We appreciate the reviewer for his valuable input. The bead-droplet interaction is indeed the core innovation of the study and the foundation on which the droplet microfluidic implementation of solid-phase synthesis is based. An extensive discussion of the design principles will help its adoption for a plethora of other applications. While the impact of the change in critical parameters was summarized in the initial manuscript, a more elaborate discussion was included in the Supplementary Information as pointed out by the reviewer.

Action

The following actions were taken in line with the reviewer’s suggestion:

- Going with the reviewer’s excellent suggestion, we have provided a more elaborate discussion of the impact of critical parameters in **paragraphs 3-5 of the Physical Working Principle of DBDR section of the main text**. The effect of droplet size on the encapsulation and ejection process, the choice of using dielectrophoretic force on the droplet to drive the encapsulation and ejection process, the alignment of the electrode with respect to the droplet generator channel, the choice of the silicone oil suspension medium and the surfactant are all explained in detail based on material and geometrical parameters.
- We moved **Fig. S4 to the main text as Fig. 3**. In Fig. 3 we have also included additional droplet size dependent experimental images of the physical process to substantiate the numerical simulations.

The numerical and accompanying experimental results for the droplet size dependent behavior of the encapsulation and ejection process are shown below for the larger droplet (Review Response Figure 1(a)) and the smaller droplet (Review Response Figure 1(b)). The dielectrophoretic force depends on the particle size and supply voltage across the trap electrodes ($\propto R_d^3 |V_s|^2$). So, the larger droplet ($R_d = 50\mu m$) can overcome the capillary force to encapsulate the bead at a much smaller voltage ($V_s \approx 85V$) compared to the nominal droplet ($R_d = 25\mu m$) which is discussed in Fig. 2 of the main text. On the other hand, the smaller droplet ($R_d = 20\mu m$) cannot completely encapsulate the bead at the same voltage ($V_s \approx 117V$) as the nominal droplet. It requires a higher voltage ($V_s \approx 135V$) to exert sufficient dielectrophoretic force to overcome the capillary force and encapsulate the bead as is evident from the simulations in Review Response Figure 1(b).

Review Response Figure 1. (a) Complete encapsulation of bead into larger droplet ($R_d = 50\mu m$) at a smaller supply voltage ($V_s = 85V$) (b) The bead is not encapsulated into the smaller droplet ($R_d = 20\mu m$) at $V_s = 117V$ which suffices to encapsulate the bead into a droplet with $R_d = 25\mu m$. The smaller droplet requires a larger voltage ($V_s = 135V$) to encapsulate the bead. The scale bar is $40\mu m$.

Question 3

Fig 3 shows the change in the fluorescence signal compared to the control group, and its reaction principle has already been mentioned in Fig 1. Please consider merging Fig 3 with Fig 4.

Our Response

Explanation

We agree with the reviewer's suggestion that Fig. 3 (demonstration of solid-phase synthesis in DBDR) and Fig. 4 (comparison of reaction fidelity in DBDR with synthesis columns) of the original manuscript should be merged into a single figure that highlights all our experimental results of the reaction chemistry.

Action

In the revised manuscript, we have merged the demonstration of the solid-phase synthesis using DBDR (Fig. 3) and its comparison with column synthesis (Fig. 4) can be merged into a single figure. **In the revised manuscript it is Fig. 4.**

Question 4

Can multiple rounds of nucleotide synthesis be achieved using the current DBDR system? Also, how to recover the bead with the nucleotide sequence from the many unreacted polyethene particles intelligently? It seems to be easily lost.

Our Response

Explanation

The reviewer is asking a very relevant question. In this context, I would first like to clarify that the manuscript focuses on achieving the following three objectives:

1. Overcome the limitations of current trapping and micromanipulation techniques that maneuver microscopic particles in a single fluidic suspension medium to establish DBDR as a reliable physical mechanism to encapsulate and eject individual beads from microdroplets by overcoming the capillary force to move microparticles across the micron sized interfaces of immiscible fluids.
2. Use this physical process to achieve the droplet microfluidic implementation of solid-phase synthesis demonstrated through the enzymatic coupling of a single nucleotide to the surface of a bead functionalized with appropriate initiator strands. Implementation of solid-phase DNA synthesis in droplet microfluidic systems has been a long-standing challenge in synthetic biology.
3. Demonstrate improved interfacing of reagents with solid supports achieved using DBDR compared to traditional synthesis columns as a route to achieve enhanced reaction fidelities.

Video V4 shows that a single bead can in principle be encapsulated multiple times necessary for multistep synthesis. However, in this proof-of-concept demonstration it has been encapsulated into the same droplet. This can in principle be extended to a highly parallelizable multicycle synthesis platform for the high throughput synthesis of arbitrarily long oligos by developing suitable architecture for parallel dielectrophoretic manipulation and parallel droplet generation into the reaction chamber backed by automation that could track trajectory of individual beads (solid supports) through sequence of reagent droplets. **This will address the questions of multiple rounds of synthesis and reliable bead recovery that the reviewer rightfully identifies.** However, this is a separate problem of platform architecture and not developing the core principles which will address the long-standing problem of reagent interfacing at its foundations.

Action

We have taken the following steps to address the reviewer's concerns:

- We are including a video (**Video V4**) for the reviewer's reference which shows a bead being encapsulated and ejected out of the droplet twice. This emphasizes the fact that a bead can undergo multiple encapsulations and ejections necessary for multistep synthesis reactions. This video shows that the time for which the bead remains encapsulated (T_{ENC}) can be controlled by controlling the time for which the voltage remains high (Fig. 2a of main text). **Video V4 has been discussed at the end of the second paragraph of the physical working principle of DBDR section of the main text.**
- To make it more obvious that the focus of the manuscript addresses the core problem of reaction fidelity, **we clearly state in the first paragraph of the introduction in the revised manuscript** that the overall yield (η) of long oligos (with strand length n) is related to the reaction error rate (ξ) through the equation $\eta = (1 - \xi)^n$. We then clearly state that minimizing ξ is the core problem we seek to address. The discussion in the manuscript is focused entirely on ξ .
- The **abstract** has also been modified to clarify that DBDR seeks to develop the fundamental physical process that will enable a droplet microfluidic implementation of solid-phase synthesis which in turn will address the problem of efficient interfacing of solid supports with reagents.
- In the **second paragraph of the conclusion section**, we talk about suitable parallelizable architecture to exploit the fundamental findings of this manuscript to build parallelizable platform for the high-fidelity solid-phase synthesis of arbitrarily long strands of oligos. We have cited references to prior work that have demonstrated parallelized dielectrophoretic manipulation (reference 75) and parallel droplet generation using step emulsification (the droplet generation technique used in this manuscript) (reference 73 and 74).

Question 5

What will happen if two or more beads are in the reaction area facing a droplet? It should be a frequently encountered situation in experiments.

Our Response

Explanation

The reviewer has asked a pertinent question. We indeed encountered similar situations during our many initial experimental trials to achieve optimal working of the physical process. In the process of suspending the beads in silicone oil, sometimes the beads remain agglomerated. We have encountered situations in which a two-bead agglomerated particle was encapsulated into a droplet (Review Response Figure 2(a)). In another situation, an agglomeration of multiple beads could not be encapsulated within a droplet that already had another bead encapsulated within it (Review Response Figure 2(b)). The agglomeration effectively behaved as a much bigger bead which required a higher voltage to be encapsulated. In yet another situation involving higher concentration of beads in the oil suspension, multiple single beads were encapsulated within the droplet. They were ejected separately one at a time from the droplet (Review Response Figure 2(c)). These issues can be avoided by keeping the bead concentration in the oil low enough to

ensure only one bead floats in the vicinity of the electrode and is dielectrophoretically trapped on it and there is no agglomeration of beads. The agglomeration of beads can limit optimal interfacing of reagents with solid supports. Ejection of beads at different times can also hamper the detection of reactions and interpretation of reaction fidelity. Therefore, we did all our chemistry after optimizing the bead concentrations to address these issues during our initial physical experimentation. So, the main manuscript discusses instances of single beads interacting with droplets. Nevertheless, we encountered many instances of multibead systems interacting with droplets which can have many varied outcomes that may require further studies.

Action

- We are providing additional images (Review Response Figure 2) to demonstrate instances of multiple bead encapsulation and ejection that we encountered during our device optimization process.
- The possible problems associated with the physical process and reaction detection due to the multibead systems are mentioned under **the suspending-initiated beads in oil solution heading of the sample preparation subsection of the methods section.**

Review Response Figure 2. (a) Encapsulation of an agglomeration of two beads into a droplet. (b) Inability to completely encapsulate an agglomeration of multiple beads into a droplet already consisting of a single encapsulated bead even at the maximum accessible voltage ($V_s = 200V$). (c) Ejection of one bead at a time from a droplet.

Reviewer-2

We thank the reviewer for acknowledging the achievements and significance of our work. We sincerely appreciate the time and effort put into understanding the details of our work. The

reviewer has rightly emphasized all the major accomplishments of our research that we wanted to highlight.

Reviewer-3

We are delighted to know that the reviewer feels that we have done something cool and interesting. However, he/she finds the presentation of our results and the supporting evidence confusing that requires significant rephrasing. We acknowledge the reviewer's concerns and are thankful to the reviewer for giving us very detailed feedback. This has provided us an opportunity to clearly present our results and supporting evidence in the revised manuscript.

Before delving into the specific redressals as per the reviewer's comments, we would like to take this opportunity to highlight the goals and achievements of our work and explain how the results aptly support our claims.

1. The Physical and Engineering Novelty

The key challenge in moving a microbead in and out of a microdroplet is to balance the various forces at play in the process. Unlike conventional trapping and micromanipulation which involves maneuvering of microparticles in a single fluidic suspension medium against the viscous drag force, our research involves the manipulation of microparticles (beads) across interfaces of immiscible fluids (the silicone oil suspension medium and the aqueous reagent microdroplet) against the capillary force. The trapping forces, drag forces and capillary forces follow different scaling laws (proportional to R_b^3 , $R_b v$, R_b respectively) with respect to the microparticle radius (R_b) and velocity (v). Therefore, precise engineering of the balance between the forces necessary to enable the encapsulation and ejection process requires careful selection of microparticle, droplet and electrode dimensions, material, and interfacial properties. Hence, achieving the process of encapsulation and ejection was the first key achievement of our research. Furthermore, this was achieved within a droplet microfluidic device that consists of a droplet generator dispensing a single droplet on-demand into the reaction chamber. So, the entire process of droplet generation as well as bead and droplet manipulation are performed on an integrated lab-on-a-chip environment. This integration involved additional design challenges of precisely aligning the droplet generator with the trap electrodes during device fabrication. **The entire physical process is elucidated in Fig. 2 of the main text of the revised manuscript along with the schematics in Fig. 1(a) and (b).**

2. Solid-phase synthesis in droplet microfluidics

Achieving solid-phase synthesis in a droplet microfluidic device has for long been a challenge due to the inability to controllably manipulate beads and droplet simultaneously in a manner to interface the mobile reagents within the droplets with the immobile reagents bound to the bead (solid support). The physical process of bead encapsulation and ejection overcomes this long-standing challenge. We use it to demonstrate the enzymatic coupling of nucleotides onto solid

supports functionalized with appropriate initiator strands as a proof-of-concept of the viability of the platform for solid-phase DNA synthesis. The droplet consists of a fluorescently labelled (with Alexa Fluor 647) single nucleotides and the enzyme TdT (terminal deoxynucleotidyl transferase) in a reaction buffer. The bead has a single nucleotide strand terminating with a 3'-OH that is accessible for enzymatic coupling of the incoming nucleotide. When the bead is encapsulated within the droplet the reaction takes place. The enzyme TdT catalyzes the binding of the fluorescently labelled nucleotides to the 3'-OH of the initiator strands. The presence of red fluorescence (due to Alexa Fluor 647 tag of the nucleotides) on the bead after ejection from the droplet indicates a successful reaction. This process was repeated three times, and a successful reaction was detected in each case. Furthermore, to eliminate the possibility of non-specific binding of the fluorescently labelled nucleotides to the 3' end, a control experiment was performed in which beads without initiator strands were encapsulated and ejected from the reagent droplet devoid of the enzyme. The absence of red fluorescence on the bead indicated the absence of non-specific binding of nucleotides to the bead surface. **This is elucidated in Fig. 4(a) and (b) of the main text in the revised manuscript along with Fig. 1(a) and (d) of the main text.**

3. Reaction Fidelity

A critical problem in traditional column-based solid-phase synthesis is the formation of deletion byproducts due to reduced reaction fidelities. Inadequate reagent access to the surface of the beads which are tightly packed in the synthesis columns leads to reaction inefficiencies. As a result, when beads functionalized with initiator strands (identical to DBDR) are flushed with synthesis reagents (identical to the droplet in DBDR), reduced fluorescence intensity is observed over a large fraction of beads. This confirms a corresponding reduction in reaction fidelities. **The operation of the synthesis column is shown in Fig. 4(c).** To further establish the reduction in fidelity is due to the physical problem of inadequate reagent access to the bead surfaces, we resorted to the modelling of the physical process of interfacing of mobile reagents (fluorescently labelled nucleotides and enzyme) suspended in the reaction buffer with the immobile reagents. In the **second paragraph of the coupling fidelity enhancement section**, we discuss this in detail in terms of the access to reagents between the pores of the beads when stacked in synthesis columns. This is depicted in **Fig. 5 of the main text** as well in the revised manuscript. On the other hand, beads reacted using DBDR end up yielding fluorescence intensities close to the maximum intensities attained using synthesis columns. The reduction in fluorescence intensities on an overwhelming majority of the beads significantly reduces the reaction fidelities in synthesis columns. DBDR leverages the power of micromanipulation to precisely interface microbeads (solid supports) with reagent microdroplets for optimal reagent access.

We have also shortened our references into a more concise and focused list by reducing more than 25% of the referenced from the initial submission as per the reviewer's suggestions. The current reference list would be necessary to provide the readers with a window to the entire spread of concepts touched upon in the manuscript.

The conclusion and outlook section have been merged into a single conclusion section.

Introduction

Question 1

Too long – a 3-page introduction. Cut it down by at least half.

Our Response

Explanation

As per one of the reviewer's later suggestions, we have clearly segregated the achievements and future potentials of our manuscript and moved the future potentials to the conclusions and outlook section at the end of the manuscript. This allowed us to reduce the length of the introduction.

Action

In the revised manuscript, the introduction has been reduced by half to approximately 1.5 pages from 3 pages.

Question 2

Apart from the increase in the need for more reagents and solvent waste, the disadvantages listed for column based solid-phase synthesis are also limitations of microfluidic-cum-electrophoretic device manipulation synthesis (e.g. labor and time intensive sample handling, complexity of automated systems – your method is far from automated, facile or speedy) – be true about the advantages of what you have done.

Our Response

Explanation

We agree with the reviewer that our platform in its current form has many of the disadvantages of column-based synthesizers. However, proof-of-concept droplet microfluidic implementation of solid-phase synthesis introduced in this manuscript has the potential for parallel, multistep, and high-throughput operations when scaled up using suitable architecture. That is within the scope of future work and discussed in the conclusion of the manuscript. Our modified and shortened introduction in the revised manuscript has more concise claims of novelties, achievements and advantages of the system in its current form as stated by the reviewer.

Action

We have moved the potential advantages of a completely automated, high-throughput and massively parallel droplet microfluidics based solid-phase synthesis platform based on our proof-of-concept demonstration to the end of the **second paragraph of the conclusion section** in the revised manuscript. The **introduction** in the revised manuscript only focusses on the novelties and achievements of the presented droplet approach and its advantages in its current state of development compared to synthesis columns.

Question 3

Your method has disadvantages – you only couple 1 thing! You cannot sell that it can be used to couple long oligomers as many times as you do. You can mention that it could be developed to allow this in the future, and the high success rate of the coupling reaction will facilitate this, but now this is not something that should be repeatedly mentioned throughout the manuscript – it is massively overpromising something that has not been demonstrated.

Our Response

Explanation

We agree with the reviewer that ours is a proof-of-concept work that shows a single reaction. However, synthesis inefficiencies and resulting lower product yield of long strands of oligonucleotides arise largely due to the inefficiencies (leading to errors) associated with single reactions which accumulate over multiple synthesis cycles. Our numerical models together with fluorescence measurements indicate inadequate reagent access to the solid supports due to bead-bead stacking in synthesis columns as the physical source of errors in individual chemical reactions implemented in synthesis columns. Our fluorescence measurements on this single reaction indicate that coupling fidelities more than synthesis columns can be achieved using DBDR. This makes DBDR a potentially viable approach for the synthesis of longer strands. We agree with the reviewer that this is a potential which requires further architectural level device development that involves scaling up of the basic concept of DBDR demonstrated here. In fact, we have clarified this in the introduction of our response to the reviewer. We understand that including the potential in the introduction may confuse the readers to think that those results have already been achieved. Nevertheless, the proof-of-concept result is still significant as it introduces a fundamentally novel approach to implement solid-phase synthesis that addresses the issue of suboptimal interfacing of solid supports and reagents in synthesis columns which is at the core of observed reaction errors in synthesis columns that necessitates excess reagent consumption and wastage to overcome.

Action

The following are the actions taken to address the reviewer's concern.

- In the **first paragraph of the revised introduction**, we emphasize how the inefficiencies in single reactions accumulate over multiple synthesis cycles to ultimately curtail the yield of long strands of oligonucleotides. So, addressing inefficiencies over individual reactions is unavoidable to achieve practical yields of long strands of oligonucleotides (although not the only requirement). To make it easy for the reviewer and the general reader to understand the focus of the manuscript, we have explicitly stated the equation for the synthesis yield $\eta = (1 - \xi)^n$ in terms of the reaction error rate (reaction inefficiency) and synthesis strand length (n) in the first paragraph of the introduction. Further, we have clearly stated that the focus of this manuscript is to address the underlying physical problems that lead to inefficiencies (large ξ) associated with solid-phase synthesis reactions. We have explicitly stated this.

- The focus of the manuscript on addressing the physical issue of inadequate interfacing of reagent and solid supports that lead to reduced reaction fidelities is clarified in the **abstract** as well.
- We have curtailed our optimistic claims of the future potential of the device **throughout the manuscript**.
- We have moved the discussion of the potential of the proof-of-concept demonstration for high-throughput multistep synthesis of long oligonucleotides (large n) to the **end of the second paragraph of the conclusion section**.

Question 4

Other disadvantages are that at no point do you recover the coupled material – so this can only be used for applications where the materials can remain attached to the bead – do you even recover the beads? This was not clear.

Our Response

Explanation

The reviewer has asked a couple of pertinent questions. We have clearly stated the objective and achievements of our work in the introduction to our response to the reviewer. While we agree that we have not recovered the synthesized material from the solid support, that does not take away the novelties and the core achievements of our work which address important long-standing problems of inappropriate interfacing of reagents and solid supports concerning current state-of-the-art synthesis platforms using a novel approach of micromanipulation based precise interfacing of microbeads (solid supports) and microdroplets (tiny packets of reagents). Our proof-of-concept results indicate that DBDR has the potential to address the fundamental issue of synthesis fidelity arising due to reagent solid-support interfacing at the core of solid-phase synthesis technologies. As device architecture is developed to scale up the proof-of-concept development presented in this manuscript subsequent integration challenges such as the one posed by the reviewer will be taken care of. This manuscript on the other hand deals with the core principle of synthesis fidelity that effects individual reactions and accumulates over multiple synthesis cycles. It introduces a novel physical process that can address this long-standing challenge at a fundamental level. This may still be useful for applications like DNA data storage where the intended goal is to achieve high fidelity (minimum error) synthesis of oligo sequences onto solid supports as bits of digital data and not to recover and use them for subsequent biological experiments.

We also want to confirm that the bead is not recovered from the droplet microfluidic device in our experiments. This is in line with the eventual aim to achieve a droplet microfluidic implementation of the design-build-test cycle in synthetic biology. While every other step of this cyclic process has a droplet microfluidic implementation, a droplet microfluidic analogue for solid-phase oligonucleotide synthesis does not exist. This makes our proof-of-concept results extremely significant. However, the bead is extracted out of the reagent droplet inside the device. This makes the analysis of the reaction products possible.

Action

The following additions are made to the **end of the second paragraph of the conclusion** section taken to address the reviewer concerns:

- We mention that DBDR can be useful for applications like DNA based digital data storage where the primary requirement is reaction fidelity. There is no need to extract the synthesis products from the bead.
- DBDR can be combined with existing droplet microfluidic implementation of other steps of the design-build-test-learn cycle of synthetic biology such as gene assembly, transformation, cell culture and colony picking to design an entirely droplet microfluidic process pipeline for the iterative process to genetically engineer biological systems. This will significantly curtail reagent utilization in this otherwise resource intensive process.

Question 5

It is not high throughput or parallelizable in this form, here the proof of concept is applicable – could this be developed for a parallelizable coupling or multistep process – move to conclusions and outlook from the intro.

Our Response

Explanation

We agree with the reviewer that high-throughput and parallelizable operation is a part of the architectural development of a practical device which belongs to the future scope of research. We have clearly stated this in the initial detailed discussion explaining our work to the reviewer.

Action

In the revised manuscript, we have moved the discussion regarding high-throughput and parallelizable capabilities to the **end of the second paragraph of the conclusion** section.

Device and Reaction Design

Question 1

This contains little information for the length. Again, it is too long.

Our Response

Explanation

The two paragraphs in this section explain the device and reaction design respectively. In the first paragraph, we discuss the choice of materials and relevant dimensions of all the device components and provide reasons for those choices. In the second paragraph, we discuss the way solid-phase synthesis was set up in DBDR for detection and reaction analysis. These discussions will help readers extrapolate our proof-of-concept work to meet the requirements of their specific end

application. We agree that we had missed some essential information in the first paragraph which we have rectified in the revised manuscript. However, we believe the length of the section is needed to convey all the detailed information regarding system design. The more specific concerns of the reviewer with respect to this section are addressed in response to his following questions.

Action

The following actions are changes are incorporated to address the reviewers' general concerns regarding lack of information in the section:

- In the revised manuscript, the actual dimension of the fluid filled reaction chamber ($3\text{ mm} \times 10\text{ mm} \times 0.2\text{ mm}$) was provided in the **third line of the first paragraph of the device and reaction design** section.
- The width ($\approx 25\ \mu\text{m}$) and height ($\approx 20\ \mu\text{m}$) of the droplet generator channel are provided separately along with a reference to Extended Data Fig. E1(a)-(c) which provide profilometric data about the shape and dimensions of the microchannel and the reaction chamber. This information can be found in the **fourth line of the first paragraph of the device and reaction design** section.

Question 2

How were the flow conditions optimized? What were they optimized to? Did this vary between devices? Operating range / values of flow rates, diagram of nozzle dimensions, chip height – is the chip $15\ \mu\text{m}$ high, but the droplets have a $25\ \mu\text{m}$ diameter, so the droplets are not spheres but are disks – was this allowed for in the volume calculations?

Our Response

We appreciate the reviewer's attention to details. Clarifying his/her concerns will help readers better understand the functioning of our device.

Explanation

We would like to explain the two primary queries of flow rate optimization and device dimensions relative to the droplet size raised by the reviewer separately.

- We would like to clarify that the fluid flow in the microchannel was driven by piezoelectric pressure controllers. Its fast-switching response facilitates the generation of a single droplet. So, the droplet generation was driven by a pressure pulse and not by fluid flow rates. A pressure of approximately $\approx 100\text{ mbar}$ was used to drive fluid flow into the microchannel and an additional $\approx 10\text{ mbar}$ pressure pulse for $\approx 300\text{ ms}$ was used to dispense a single droplet into the reaction chamber. Maintaining the high pressure for a longer duration or using a higher pressure would result in the generation of multiple droplets. So, the pressure pulse was optimized for single droplet generation. The same was used across all devices. We discussed these additional details with figures in the supplementary information of the revised manuscript. The 100 mbar steady pressure

depends on the height of the device relative to the fluidic tank (a picture of which has been provided in Extended Data Fig. E2(d)). It therefore is a function of the experimental setup and could be modified. At the same time, it is sensitive to the tightness of the fluidic connections, the precise amount of reagent in the input tubing, and the precise dimensions of the fluidic channel. This leads to variations across experimental trials and devices.

- The microchannel is $\approx 25 \mu\text{m}$ wide and $\approx 20 \mu\text{m}$ high whereas the reaction chamber is more than 3 mm wide, 10 mm long and $200 \mu\text{m}$ high. So, the droplet that has a $50 \mu\text{m}$ diameter (not $25 \mu\text{m}$ diameter as the reviewer points out) maintains its near spherical shape while forming nearly 140° contact angle on the bottom glass surface of the device. This is depicted appropriately in Fig. 1(a) where the droplet, bead and the electrodes can be seen enclosed within the fluid filled reaction chamber. This explains the significance of the schematic. We also hope this addresses the reviewer's concerns regarding the volume calculations of the droplet.

Actions

- In the revised manuscript, we have provided additional details of fluid flow in the microchannel leading to the droplet generation in the first paragraph of **device and reactor design** and in the first paragraph of the **physical working principle of DBDR** sections of the revised manuscript. Specifically, we mention that a piezo driven pressure controller is used to drive droplet generation. A single pressure pulse of 10 mbar for $\approx 300 \text{ ms}$ is used to dispense a single droplet into the reaction chamber. Video V5 shows a single droplet generation when such a pressure pulse is used (Review Response Figure 3(a)). We also highlight here how multiple droplets are generated (Video V6) in response to a higher magnitude pulse (Review Response Figure 3(b)). In our experiments we wanted to avoid these multiple droplet generation scenarios. These additional details are provided in the extended data of the revised submission. **Review Response Fig. 3(a) and (b) are included into Extended Data as Figure E4(a) and (b).**
- We have provided additional profilometric depth data to go with the optical images that provide the height measurements near the intersection of the transition region between the microchannel and reaction chamber as additional information in the **Extended Data Fig.E1 to clarify that the droplet is generated by a $20 \mu\text{m}$ channel (Reviewer Response Figure 3(c))**. However, once generated it sits in the $200 \mu\text{m}$ reaction chamber. Therefore, the reaction chamber is sufficiently large for the droplet to maintain its spherical shape without being distorted. The dimensions of the reaction chamber are included in the third line of the first paragraph of the device and reaction design section of the main text.

Question 3

Supp F1 – actual picture of the diagram next to the diagram.

Our Response

Explanation

Review Response Figure 3. (a) A 10 mbar and ≈ 300 ms pulse is used to generate a single droplet. (b) A 20 mbar pulse of approximately the same duration generates two droplets. (c) Dimensions of microfluidic droplet generator channel and chamber are given by the blue and red plots respectively. The blue and red plots are taken along the blue and red lines shown in the inset of the microfluidic device image.

The reviewer makes a very valid point here. Supplementary Figure S1 in the original manuscript represents the COMSOL Multiphysics simulation geometry setup used to model the process of encapsulation and ejection. As the label indicates, it is an axis symmetric simulation. Its relation to the simulation color plots presented in Fig. 2€ is not obvious.

Action

For the purpose of added clarity, we take the simulation color plots in Fig. 2(e) and add it to **Supplementary Figure S1** in the revised manuscript to clearly explain how the simulation plots presented in the main text are related to the simulation setup.

Physical Working Principles of DBDR

Question 1

Too long (2 ½ pages). Condense this down into salient information (1-2 paragraphs) and expand the supplementary information on this. I am not a mathematician so am not able to comment on

these aspects, but it seems that a lot of the information in this section is repeated in the supplementary information. Please still include it, but summarize it in the main paper, and expand the supplement for the interested reader.

Our Response

We respectfully disagree with the reviewer on this point. Our key achievement is the development of the physical process that enables DBDR. Hence, it is necessary to explain the novelty, the challenges and the achievements of the physical process in great detail. This has been rightfully acknowledged by the other two reviewers in their comments. In fact, reviewer 1 wants us to strengthen certain aspects of the discussion with respect to the physical process. So, we believe a detailed discussion of the working mechanism of DBDR deserves a place in the main text of the manuscript. The mathematical nature of the discussion provides the necessary scientific accuracy to the claims and helps understand the process of bead encapsulation and ejection in terms of the particle dimension dependent scaling of the dielectrophoretic and capillary forces. The supplementary information provides additional details about how the mathematical modelling was set up to solve for the two-phase fluid flow under the influence of electric field. This is represented by the two dimensional red-blue plots in Fig. 2e(ii). In this plot, blue represents the oil suspension medium while red represents the aqueous droplet. The transition from blue to red represents the interface of the droplet and the medium. This section also provides details of a more intuitive description of the same process in terms of the total electric energy and Gibbs free energy of the system. This is represented by the encapsulation and ejection plots in Fig. 2e(iii) and (iv). In the main text, the inferences drawn from those detailed mathematical calculations are used to explain the working of the encapsulation and ejection process. Further reducing the description of the physical working principle will understate the importance of the novel physical process to the entire work presented in the manuscript.

We do realize that the section is complex due to its technical content. In the revised manuscript, we have tried our best to make the writing clear while maintaining the critical technical details.

Enzymatic Coupling of Nucleotides in DBDR

Question 1

How did you choose this optimal droplet size? You give an overly long explanation of this without showing the evidence of “failed” experiments that you used to reach the optimized conditions – these optimizations should be included in the supplementary information.

Response

Explanation

The choice of droplet size is based on physical considerations. A large droplet size would naturally entail the consumption and wastage of more reagents per bead. So, ideally the smaller the droplet sizes the better it is. However, as the droplet size decreases, the dielectrophoretic force ($\propto R_d^3 |V_s|^2$)

required to overcome the capillary force and encapsulate the bead inside the droplet increases. This would require a larger voltage (V_s) to encapsulate the bead into a smaller droplet. We have shown this using numerical simulations and experimental snapshots in the modified manuscript. Even if a higher voltage was accessible, the dielectric breakdown field strength of the silicone oil suspension medium would ultimately define the smallest droplet in which the bead can be encapsulated. Hope this helps the reviewer understand the physical basis of the choice of droplet size.

Action

In the revised manuscript, we have provided a short and precise discussion on the choice of optimal droplet size in the third paragraph of physical working principle of DBDR. This discussion is further strengthened by the **introduction of Fig. 3 from the supplementary information into the main text** on the recommendation of reviewer 1. Fig. 3a and b combine to emphasize how it was difficult to encapsulate beads into smaller droplets. This is an example of failed experiments in which we could not load beads into droplets of decreasing sizes which would have made the reactions faster and interaction of reagents with the solid support more efficient. So, we chose the optimal size of $R_d = 25 \mu m$ while limiting the supply voltage to 120 V.

Question 2

“triplicated” – do you mean that you injected, reacted, and retrieved a bead 3 times in 3 separate droplets and then averaged these fluorescent image results? Or did you do it loads of times and pick the best 3? In either case, this is not enough time to jump to the conclusions you have from here. If it's more, show the data and how you calculated the values and error in the values.

Our Response

Explanation

We have clarified this in the revised submission. In our experiments, a successful reaction in the presence of all reaction ingredients was used to demonstrate the enzymatic coupling of oligonucleotides onto the initiator strands immobilized onto the beads. We replicated the experiment three times to ensure reproducibility. A control experiment is implemented using beads devoid of initiator strands and droplets devoid of enzymes. As expected, there was no coupling of fluorescently labelled nucleotides onto the bead surface. This was inferred from the absence of fluorescence from the bead surface that was ejected from the droplet. In the context of this question, we would like to compare our work to that of another apparently similar work by Liu et al. that was brought up by the reviewer in a later question of the review. In their work, unlike ours, the enzyme is bound to the solid support while the reaction products are formed in the droplet. Depending on the degree of interaction with the enzymes bound to the beads, different regions of the droplet will have different amounts of product formation. Moreover, the reactor droplet in Liu et. al.'s work is $\approx 75 nl$ compared to microdroplets of $\approx 65 pl$ in our case. The three order of magnitude larger volume means an order of magnitude larger length scale. This would result in two orders of magnitude larger diffusion time in Liu et. al.'s reactor droplet ($\approx 1250 s$) compared

to our reactor droplet (≈ 12.5 s). This (≈ 1250 s) is much longer than the ≈ 115 s required for the droplet to breakup at the third break up channel. So, there is insufficient time for the reaction products to homogenize across the large reactor droplet. It is also possible that there was no catalytic activity of the enzyme bound to the bead in certain regions of the main reactor droplet due to the absence of beads in that region. If the same part breaks into the detector channel no reaction is expected to be detected. So, as the main reactor droplet breaks up into multiple smaller droplets, varying quantity of reaction products are expected across these. This requires a detailed statistical analysis over a large number of these smaller droplets to confirm reactions. On the other hand, in the case of DBDR, the reaction products are formed on the bead which is exposed to the same reagent environment irrespective of its position within the droplet. This makes it obvious that in the case of DBDR (unlike in the case of Liu et. al's work), there is no chance for a coupling reaction to not take place in the presence of all the appropriate ingredients. So, we believe a large data set is not required to establish our claims of a successful reaction. We have cited a couple of high impact publications as references which have used just one fluorescence measurement to confirm the occurrence of coupling reaction in solid-phase synthesis.

Action

The following actions were taken to clarify the concerns of the reviewer:

- We added a small comparative discussion between our work and that of Liu et al **at the end of the first paragraph of the conclusion section** of the revised manuscript explaining the important differences.
- We cited two high impact publications as references (reference 63 and 64 in the **17th line of the enzymatic coupling of nucleotide in DBDR** section of the revised manuscript) which have reported the use of a similar fluorescence-based approach to establish the occurrence of reactions without multiple repeats. Our approach followed this precedence.

Question-3

Explain how challenging this technique is by showing what failure looks like, and being honest about how much skill and effort it takes to do this properly, and how you tell if it worked or not. For example, in AFM protein folding and unfolding there are thousands of events recorded, of which only a small percentage show folding / unfolding trace. The only reason this technique has value is because those that developed it were honest about the failure rate and how to do statistically relevant repeats to get the information from the technique. I cannot gauge this information from this manuscript, and this makes me skeptical.

Our Response

Explanation

The reviewer has raised a very valid point. The physical process of encapsulation and ejection of the bead from the droplet is indeed a very challenging process that required skillful device design, fabrication, and experimentation. We have highlighted these challenges in the revised manuscript.

The device design involved the choice of droplet and bead sizes based on accessible voltages and breakdown field strength of silicone oil. This led to the chosen dimension of the microchannel and the electrode. We have summarized this in the first paragraph of **device and reactor design** section and the 2nd to 4th paragraph of the **physical working principle of DBDR** section. If a smaller gap between the electrodes were to be chosen, there would not be enough space for the droplet to withdraw from the bead leading to incomplete ejection (Review Response Figure 4(a)).

The experimentation also involved multiple challenges. To ensure desired interplay of dielectrophoretic and capillary forces to drive the encapsulation and ejection process, precise control over the piezo driven pressure controller is important. However, due to the sensitivity of the steady state pressure to multiple factors such as tightness of fluidic connections, height of fluids in the inlet tubing the steady state pressure fluctuates. Sometimes, it is higher than the exerted pressure using the piezo driven pressure controller. This drives fluid to flow back into the channel from the chamber. In this case, the interacting beads and droplets within the reaction chamber are drawn back into the droplet generator channel (Review Response Figure 4(b)) due to the dominant viscous drag force. Some other times, the steady state pressure is lower than the exerted pressure. In such cases, the pressure pulse used to generate droplets would be higher than necessary to generate a single droplet, multiple droplets would be dispensed into the reaction chamber (as discussed in Review Response Figure 3). In such a case, when a large voltage is applied to drive bead encapsulation into a droplet, it is the droplet-droplet merger that would be the dominant effect (Review Response Figure 4(c)). This is due to the much larger size and dielectric constant of the droplets compared to the suspension medium which lends it a stronger electrical response to the applied A.C. voltage.

The device fabrication process also offered many challenges. The high voltage required to drive anodic bonding of the silicon and glass wafers many times led to rough electrodes. Sometimes droplets moving away from the electrodes during bead ejection get stuck on the rough electrode surfaces. This leads to their breakup into a tiny daughter droplet which sticks to the bead. This prevents reliable detection of enzymatic coupling. So, devices with such fabrication inefficiencies were discarded from performing chemical reactions.

These challenges are the source of failures in the device operation and make it challenging to get a successful experimental repeat. However, every time the physical process has worked properly, we have been able to achieve successful coupling reactions or control experiments. We agree that it would be necessary to improve the success rate of the physical process as it is taken from proof-of-concept to the realm of practically scalable massively parallel and high-throughput microfluidic device architecture. We also believe these failures would provide useful information for researchers who intend to further develop the proof-of-concept results presented in this manuscript. We are thankful to the reviewer for bringing this up.

Actions

The following actions are taken to address the reviewer's concerns:

Review Response Figure 4. (a) Incomplete ejection of bead from droplet due to insufficient spacing between electrodes for droplet motion. The scale bar is 110 μm (b) Bead-droplet suction into the droplet generating microchannel due to low steady state pressure which drives backward fluid flow into the microchannel. This exerts a viscous drag force on the bead-droplet system to drag them into the microchannel. (c) The much larger dielectrophoretic force on droplets drives droplet mergers under the influence of electric fields rather than bead encapsulation into droplets. (d) Breakup of a bigger droplet into a smaller daughter droplet surrounding the bead during the process of ejection due to rough electrode surface.

- The effect of a variations in pressure leading to the generation of multiple droplets into the reaction chamber or the suction of beads and droplets from the reaction chamber back into the microchannel is discussed under the **encapsulation and ejection of bead from droplet heading under the experimental procedure subsection of the methods section**. **Extended Data Fig. E4** shows these pressure-dependent effects along with Video V5 and Video V6.
- The dielectrophoretic force induced merger of multiple droplets is discussed at the end of **supplementary information section 2** and in **supplementary information Fig. 2**.
- The adverse effects of fabrication imperfections on device operation are mentioned in the **device fabrication** subsection of the **methods** section.

Question-4

You don't recover your coupled molecules, so the low error rate on coupling doesn't really matter in comparison to other nucleotide coupling techniques – this issue crops up repeatedly and is mentioned at the top.

Our Response

Explanation

The reviewer rightly observes that we do not recover our coupled molecules from the device. However, we do not think that it absolutely prevents us from drawing inferences about coupling fidelities. Fluorescence measurements can be used for in situ confirmation of binding of fluorescently labelled nucleotides to the initiator (primer) strands and the relative intensity can be used as a measure of relative coupling efficiencies as long as the reaction products are spatially separated from the reagent droplet containing the uncoupled fluorescently labelled nucleotides. There is precedence to such fluorescence dependent inferences of relative binding efficiencies in the supplementary information of reference 63 cited in the revised manuscript. Moreover, even in the work of Liu et. al. (that is referred to by the reviewer in a later question) a similar strategy is followed. Fluorescence intensity in the daughter droplet is measured to infer reaction efficiency once it is spatially segregated from the bigger reagent droplet. Even in the work of Liu et. al. it is collected in yet another chip and not necessarily extracted out for other kinds of measurements. The important point though is that the product droplet is separated from the reagent droplet. This allows us to attribute the fluorescence to the reaction products. Even in our case, after ejection, the bead is separated from the reagent droplet. This allows us to use fluorescence as a tool to measure coupling efficiency. To compare with synthesis columns, we did the same reactions in them and then measured the fluorescence of the beads. **This is described in detail in the revised manuscript and the methods section.** We agree with the reviewer that if the beads are extracted from the devices, then many other well established characterization techniques can also be used to compare reaction fidelities with respect to synthesis columns. However, in situ fluorescence gives us enough information to draw conclusions for our proof-of-concept work.

Not extracting the bead after synthesis also has another important reason which we have previously provided in response to question 4 of the reviewer in the introduction section. An important end objective of our work is the design of a droplet microfluidic based design-built-test-learn cycle for synthetic biology that could minimize reagent consumption in this iterative approach to achieving optimal genetic structures for intended functions. For this purpose, the synthesized molecules need not be extracted from the droplet microfluidic device. Rather, integration of solid-phase synthesis with the subsequent step of gene assembly within the droplet microfluidic platform is required. For this, the beads with the synthesized molecules must be extracted from the reagent droplet. As we clearly show in this work, we have been able to achieve that. DBDR is also suitable for DNA based data storage where the primary requirement is high fidelity synthesis for error free writing of digital data onto solid supports. While we acknowledge that a lot more architectural developments may be needed to make the basic demonstration presented in this manuscript into scalable practical platforms as well as the integration of solid-

phase synthesis with gene assembly in a droplet microfluidic platform, our achievement of realizing a plausible route to implement solid-phase synthesis in droplet microfluidics is a fundamentally important and unavoidable first step. This is the main highlight of our work. Additionally, the high synthesis fidelity makes it a potent platform for many applications which need further exploration.

Actions

The following actions were taken in response to the reviewer's concerns:

- In the **enzymatic coupling of nucleotide using DBDR section line 17** we have cited three references (57, 63 and 64) which used fluorescence to detect chemical reactions.
- We have added additional sentences in our revised manuscript stating the need for additional development to make the device practically more viable at the **end of the second paragraph of the conclusion** section.
- In the **first paragraph of the introduction**, we clearly state using mathematical representation that we focus on reaction errors as the fundamental underlying problem and not on other aspects of the synthesis. This ensures that our claims do not exceed our demonstrated results.
- We have also clearly stated in the **abstract** that the manuscript focusses on addressing the fundamental physical problem of efficiently interfacing beads (solid supports) with reagents by developing a novel approach based on precise microscopic control of reagent microdroplets and beads.

Coupling Fidelity Enhancement in DBDR

Question-1

My main issue with this section is that there are many ways of optimizing the control process to improve the yield. For example, mixing the beads on a cheap blood wheel mixer, as is commonly used in manual solid-phase peptide synthesis, or sparging the reagents so the column is only packed upon recovery, would significantly improve mixing, access of reagents to reactant surface, and thus the fidelity of the reaction. This control is deliberately making the worst possible result using an existing technology to make the DBDR process look better. It is way too long, and because the work doesn't recover the material, they are unable to compare the coupling fidelity to published experimental techniques.

Our Response

We thank the reviewer for bringing up other methods of optimizing reaction efficiencies in column synthesis platforms, for example, the blood wheel mixer used for general agitation of reagents to improve reaction efficiencies. As per the reviewer's reference, such methods have been employed to improve efficiencies specifically in peptide synthesis; however, there are many types of solid-phase platforms that must also be considered that employ different chemistries where effectiveness of agitation may vary (as in our case, for DNA oligo synthesis as a proof-of-concept).

First, although such mixers (including benchtop shakers) may be added to select commercial column synthesizers (e.g., K&A, AB394), it is not as practical to introduce such a process to traditional fixed-column/ plate instruments (e.g., Shasta-96, 192, Mermade, Dr. Oligo) without significant re-engineering. Secondly, it is understood (I. Alshanski et. al. *Org. Process Res. Dev.* **22**, 1318-1322 (2018)) that shaking leads to limited improvement in mass transfer and diffusion reaction in heterogeneous reactions like the solid-phase synthesis where one reagent and the final product are immobilized on solid supports. This limits improvement in reaction efficiency. Moreover, this helps emphasize the novelty aspect of our platform. All existing methods to improve reaction efficiency through better interfacing of solid supports with mobile fluidic reagents using various approaches to mix reagents involve macroscale forces that drive mass transfer in the reactor. This leads to improved probability of interaction between the reacting species. On the other hand, we seek to do this by using microscale trapping and micromanipulation forces. Our platform also has the potential to scale up by combining the massive parallelizability and high throughput capabilities of droplet microfluidics and trapping and micromanipulation. This we have discussed at the **end of the second paragraph of the conclusion section.**

Question-2

For example, later they mention that microarrays are not as good at this process. LeProust et al.(2010) *Nucleic Acids Research*, Volume 38, Issue 8, Pages 2522–2540<https://doi.org/10.1093/nar/gkq163> synthesize 150mer oligos on arrays with a yield of 1 pMol full-length product with 4-5% yield from the crude synthesis by adjusting the reaction conditions to minimize detrimental side reactions. I cannot make comparisons between these works, because the current authors (1) have only demonstrated one coupling reaction and (2) have not quantified the yield in any way as they have not recovered the material. This is an example of overclaiming that needs to either be better quantified or removed completely. You have not made long oligos, so you need to dial down the number of times you say that this technique can make long oligos, because at the moment it can't.

Our Response

Explanation

The reviewer makes a valid point here. There is a lot of high impact literature on solid-phase synthesis technologies that emphasize the fact that synthesis columns have better reaction efficiencies than microarrays. In our research, we have compared synthesis columns with DBDR using fluorescence measurements. We have found that the latter yields better reaction fidelities. So, we extrapolated our interpretations to conclude that DBDR can be better than microarrays as well. We agree that there have been no direct comparisons between DBDR and microarrays in our manuscript.

Action

We have completely removed the relevant sections from the manuscript.

Question-3

Discussion of electric field surface charge enhancement of reactants at the surface is difficult to follow and seems selective and hand-wavy rather than a complete explanation.

Our Response

Explanation

We agree with the reviewer that the discussion involving the effect of electric field on the reaction in the main text of the original manuscript was not sufficiently detailed. We tried to present a very simplistic picture in the main text while leaving all the simulation details of the electric field driven migration of charged reacting species in the supplementary information. In this context, we believe that a lot more work needs to be done to understand in detail the effect of A.C. electric field on reactions in microdroplets. While there is a lot of literature on D.C. voltage (electric field) driven acceleration of reactions in microdroplets, more work needs to be done on the A.C. electric field effects. We believe such a detailed study of electric field effects would require a separate manuscript. The focus of this manuscript is to present a novel physical process which can achieve better interfacing of solid supports and reagents through precise micromanipulation. Our preliminary analysis of A.C. electric field driven migration of charged reacting species suggests that it can also have a non-negligible role in improving interfacing of solid supports with mobile reagents contained in the droplet. For the purpose of this manuscript, we just wanted to emphasize this contribution to the coupling fidelity enhancement.

Action

We move the discussion on electric field enhancement of reagent **interfacing from the main text to the supplementary information (Section-8)** as the results presented in this manuscript are not a complete discussion of the electric field induced effects on the coupling reactions. We extensively discuss numerical models that indicate improved interfacing of solid supports with mobile reagents within the droplet. In the main text, we briefly mention that the A.C. electric field may also be contributing to the enhanced interfacing of solid supports and reagents at the end of the second paragraph of the coupling fidelity enhancement of DBDR section.

Implications and Outlook

Question-1

Move the long oligo stuff. You haven't done this and a whole paragraph on it is too much.

Our Response

Explanation

As we clearly stated, our intention is not to claim more than what our results indicate. So, we accept the reviewer's suggestions.

Action

We shortened this discussion while also moving it to the **second paragraph of the conclusion section** in the revised manuscript. **The first paragraph of the conclusion section** summarizes the significance and achievements of the current work.

Question-2

Start with conclusions about what you have done, and then do the quick calculation of how this could be applied if the developments were made to allow multiple coupling reactions onto the same bead and recovery of the materials.

Our Response

Explanation

We accept the reviewer's suggestion. In response, we would like to reiterate our response to the previous concern raised by the reviewer.

Action

We move the conclusions about the significance and achievements of our work above its implications and future potential (**second paragraph of conclusion section**). In the latter section, we discuss briefly by showing calculations of yield improvements in multicycle solid-phase synthesis using DBDR over synthesis columns which give an idea about the remarkable potential of the platform.

Conclusions

Question-1

Near perfect" you haven't quantified this in any comparable way, and you haven't collected the material to measure this.

Our Response

Explanation

In the original manuscript, earlier in the section involving the discussion of coupling fidelity enhancement, we have discussed that a 3.2-fold enhancement of reaction fidelity in DBDR over synthesis columns means that even when reactions in synthesis columns are a mere 32% efficient, the reactions using DBDR can be near-perfect. So, the term near-perfect was used in the context of a comparison with synthesis column. We don't intend to claim near-perfect in an absolute sense as is apparent while reading the relevant section in the conclusion.

Action

We removed that phrase in the conclusion section to avoid similar misunderstandings.

Question-2

Fluorescence measurements suggest...” they do, lead with this, this is what you have measured.

Our Response

Explanation

As the reviewer rightly identifies, our inferences are based on fluorescence measurements, and we have clearly stated that in our concluding paragraph.

Actions

We make sure this is stated clearly in all the relevant sections of the revised manuscript.

Question-3

We have achieved the first demonstration... “ Liu et al. (2021) Biomicrofluidics, Volume 13, Issue 3,034103, DOI: 10.1063/5.0050440 use pico injection to inject beads, react them and retrieve them. You are the first to do this with an applied field though, so say that.

Our Response

Explanation

We thank the reviewer for bringing up this impressive piece of work by Liu et. al. An overview of this paper helps us identify the perceived apparent similarities between our foundational physical advancement and the previously published work by Liu et. al. that can potentially downplay and discredit the novelty and achievements of our work in the minds of the reader. However, a deeper understanding of the processes will help appreciate the important accomplishments of the novel process presented in this manuscript. So, we would like to take this opportunity to further emphasize the novel aspects of our fundamental physical process and clearly distinguish it from the work of Liu et. al.

Our Work	Liu et. al.
Bead moves across the interface of immiscible fluids i.e. silicone oil and aqueous droplets by overcoming the capillary force at the droplet-suspension medium interface. This requires the external dielectrophoretic force to counterbalance the capillary force and makes our novel process extremely challenging.	Bead pre-suspended in the aqueous phase are loaded onto the reactor droplet through merging of miscible aqueous phases. No transition across immiscible fluidic interfaces is involved. Such merger of miscible fluids for droplet loading has been extensively reported in microfluidic literature which have been cited in our manuscript (Ref. 55 and 56).
Bead is ejected out of the reactor droplet without causing generation of smaller satellite droplets. Using the hydrophobicity of the bead to eject it out of the droplet without formation	Bead remains within the reactor droplet held in place by the acoustophoretic forces. It is not ejected out of the droplet. In Ref. 55 and 56 also beads once encapsulated within the droplet are held inside it. Breakup of droplets

of satellite droplets is a difficult challenge that we could solve.	into satellite droplets at channel bifurcations as is reported in this manuscript have already been reported in many prior manuscripts. A few examples are: Chem. Eng. Sci. 188 , 158-169 (2018), Chem. Eng. Sci. 188 , 11-17 (2018).
The initiator (primer) strand is immobilized on the bead whereas the fluorescently labelled nucleotide and the enzyme are suspended in the droplet. The final product is also formed on the bead as an extension of the initiator strand.	The enzyme is bound to the bead and the reactants are suspended in the droplet. The final product is formed within the droplet.
Irrespective of the position of the bead within the droplet the initiator strand has access to the enzymes and the nucleotides. So, reaction is bound to take place and products are bound to be formed on the bead.	The reaction is more likely to take place in parts of the droplet in the vicinity of the bead which has the enzyme. Away from the bead the reaction rates will be slow in the absence of the enzyme. So, when a random section of the droplet breaks apart at the channel bifurcations, the amount of product formation is expected to vary significantly due to the dependence on the concentration of beads in that section of the droplet.
Reaction products are analyzed on the bead after separation from the reactor droplet	Reaction products are analyzed in the satellite droplet after breaking up and separation from the main reactor droplet.

Action

The following actions have been taken to emphasize the differences between our work and that of Liu et. al. to highlight our achievements:

- The physical differences between our work and that of Liu et. al are explained in **lines 24 to 31 of the 3rd paragraph of the introduction** in the revised manuscript.
- The differences in the implementation of reaction chemistry are mentioned at the **end of the first paragraph of the conclusion** in the revised manuscript.

Question-4

Vague mention of “...applications beyond solid-phase synthesis.” What are these?

Our Response

Explanation

We appreciate the reviewer for his attention to detail. We understand that just mentioning applications beyond solid-phase synthesis without precisely stating what they are seems vague. Although, we had introduced the possibility of studying enzymatic reaction in volume and surface

confined manner using DBDR in the original manuscript, it was not combined with the discussion on applications of DBDR beyond solid-phase synthesis.

Action

We rectified this in the modified submission by explicitly including it in the discussion on the applications of DBDR beyond solid-phase synthesis in the **last paragraph of the conclusion**. Specifically, we discussed the possibility of using DBDR as a route to exploit the fundamental advances made in microdroplet chemistry in solid-phase synthesis on a chip-scale droplet microfluidic platform. It can also be a great route to study enzymatic reactions in volume and surface confined environments to mimic biochemical reactions in vivo.

Question-5

Again, dial down the big claims and sell what you have actually done honestly.

Our Response

In accordance with the suggestion of the reviewer, the big claims have been significantly curtailed in the modified manuscript while clearly segregating the achievements of this work from the potential of the future.

Figure

Question-1

These are all schematics, except (c) – I had to search hard for this – move (a) and (d) to supplementary, make (c) bigger, are the arrows a calculation? Of what (the electric field)? This is not clear.

Our Response

Explanation

The 2-dimensional color plot represents the electric field magnitude, and the arrow plots represent the direction of the electric field vector. We agree that detailed explanation regarding the arrows was missing in the figure caption. We made the necessary changes in the revised submission. We hope that the reviewer will be understanding. We respectfully disagree with the reviewer's suggestion to move Fig. 1(a) and (d) to the supplementary information. We believe these schematics are necessary to explain the working mechanism of DBDR as described in the **Device and Reaction Design** section. Without proper schematics to emphasize the working of DBDR, the perceived similarities between our work and that of Liu et. al. might undermine the novelties and achievements of our work.

Action

The arrows in the 2-dimensional electric field color plot have been explained in the revised caption of Fig. 1(c).

Question-2

Are the red and blue pictures in (e) from image analysis or calculation? There is no scale on the images, is this the results of the beads not going in droplets that are too big/small? Refer to this in the text, I couldn't find this either. Put the applied field graph above the images so that you can see the image that corresponds to the applied voltage more easily.

Our Response

Explanation

The red and blue images that the reviewer is referring to are phase field plots that were obtained from coupled A.C. and two-phase fluid flow simulations using the phase field method. As the color bar below the plots clearly indicate, the blue phase ($\varphi = -1$) is the silicone oil suspension medium while the red phase ($\varphi = 1$) is the reagent phase of the droplet. The boundary is a narrow transition region between the two phases. These represent the case of a single bead being encapsulated by an optimally sized droplet ($R_d = 25 \mu m$) and ejected from it. Ideally, we wanted to put the applied field plot (Fig. 2a) on top of the other sub figures to clearly highlight the time sequence of processes. However, the different dimensions of the figures make any such regular spatial arrangement difficult. So, we have to stick to the original format and label the corresponding subfigures on the sections of the graph in Fig. 2(a).

Action

The following actions were taken to address the reviewer's concerns:

- We have added scale bar to the images in **Fig. 2(e)** to clearly represent the dimensions of the bead and the droplet. The length of the scale bar is provided in the figure caption.
- Fig. 2(e) has already been referred to in **the second paragraph of the physical working principal section** of the main text.
- The fluidic phases in terms of $\varphi = \pm 1$ have been explained in the figure caption.

Question-3

Where is the graph of the distribution that is shown for the control in F4? This is just unscaled pictures and a diagram showing what you think is happening.

Our Response

Explanation

In the revised manuscript Fig. 4a and b discuss the coupling and control experiments implemented using DBDR. Fig. 4d compare reactions implemented using DBDR with synthesis columns. So, the plots in Fig. 4d only have compared cases where actual enzymatic coupling reactions were expected to take place. Herein, we believe the reviewer has a valid point. The incorporation of the

intensity profile of the control experiment in Fig. 4b would be a great addition to the plot in Fig. 4d and present a more accurate picture of the comparison between DBDR and synthesis columns. However, this would also require the incorporation of the intensity profile of control experiments implemented using synthesis columns as well. That would make for too much data at the bottom of the plot. This would take the focus away from the comparison of actual coupling reactions.

As is clearly understood from the plot, we have used arbitrary units to define the intensity values. However, this arbitrariness is inconsequential as it is meant for comparisons between two approaches and not to establish absolute values of intensity. We want to clarify that the same arbitrary units are used for analyzing all fluorescence data. The important point though is the fact that the illumination intensity was the same for measurements involving DBDR and synthesis columns.

Action

The following changes have been incorporated in the revised manuscript to address the reviewer's concerns:

- We have also included scale bars on all the images in **Fig. 4d** to prove that these are scaled data. The details of how the plots have been generated in Fig. 4d are explained in the **Image and Data Analysis subsection of the methods section**.
- We have provided the supporting raw data of the average fluorescence intensity of beads and the code that was used to extract the bead data. Hope this will convince the reviewer that the plot uses actual data and is not a mere schematic.

Question-4

As mentioned above, there are issues with these experiments.

Our Response

We have addressed these concerns of the reviewer in response to questions 1-4 in the Enzymatic Coupling of Nucleotides in DBDR section and question 1 of the Coupling fidelity enhancement in DBDR section.

Question 5

I don't really know what F5 is adding, again there are a lot of diagrams and not a lot of data in here.

Our Response

Explanation

We would like to clarify that the role of Fig. 5 is to substantiate the experimental observation that the reactions in DBDR achieve superior interfacing of reagents and solid supports than synthesis

columns using numerical modelling of the physical processes underlying the interaction of reagents with solid supports. As has been stated previously, we explain the observed enhancement in reaction fidelity of DBDR primarily in terms of the reduction in reaction fidelity in synthesis columns due to bead-bead stacking which is avoided in our single bead-droplet reactor. This has been established through numerical simulations of particle stacking in synthesis columns under turbulent fluid flow that clearly depicts inadequate and varied exposure of beads in the stack to the mobile reagents in the solution. This is depicted in Fig. 5. Reduction of reagent diffusion time in the smaller reagent droplet compared to synthesis columns also plays a part in improving reaction fidelity in synthesis columns. We also state that the electric field driven migration of charged reacting species may play a non-negligible role in improving interfacing of reagents with solid supports.

Action

We have made the following changes to **the second paragraph of the coupling fidelity enhancement in DBDR** section:

- Made the discussion more concise and reduced its overall length to make it easier for the readers to follow the discussion.
- Focused Fig. 5 on the bead stacking simulations that reduce reagent access to the bead surfaces in synthesis columns. The 3D and the corresponding zoomed in 2D panels in Fig. 5 are plots of particle distribution in the synthesis column under the influence of fluid flow during reagent influx in the synthesis columns. These are not mere schematics. The details of the simulations leading to these plots have been mentioned in **supplementary information**. Furthermore, as a part of data and code availability requirement we have provided the code along with the input data that led to the generation of the 3D plot. The plot with appropriate scale is provided as Review Response Fig. 5.
- Moved the details of the discussion on the contribution of diffusion and electric field to enhanced reaction fidelity in DBDR to the **supplementary information** while briefly mentioning about them in the main text.

Review Response Fig. 5. Example plot of 3D particle distribution image shown in Fig. 5 of the main text. In the main text the axis has been removed from the plot.

REVIEWERS' COMMENTS

Reviewer #1 (Remarks to the Author):

The authors have satisfactorily addressed all the concerns raised. It's acceptable for publication in Nature Communication now.

Reviewer #3 (Remarks to the Author):

Thank you to the authors for carefully considering and addressing the questions regarding their manuscript. I feel that it has significantly improved the manuscript, and much better represents the excellent work they have done in a much clearer and more concise way.

The following are mainly typos, and hopefully should be much easier to address for the authors:

In some places you have a space between a number and the units, and in others the space is missing. E.g. 5.5nM/m line 142 pg 3, please check for consistency in spacing between values and units (degrees symbol should not have a space as shown on line 143, but most other cases should).

Pg 1 line 35 – remove “)” after, “initiator primer”

Pg 2 line 46 – change “smaller” to “small” and consider specifying a size range in either volume or diameter, and / or also consider specifying the size of the traditional reaction chambers as well to make the comparison clearer.

Pg 2 line 50 – “allows to dispense a single droplet...” is not great grammar, suggest changing to “where a single droplet can be dispensed...” or something similar.

Pg 2 line 54 – “towards droplet reactors...” would read better as “towards developing / using droplet reactors...”

Pg 2 line 60-64 is currently 1 sentence. This would read better if the colon was a full stop followed by “This is...”

Pg 2 line 66 – single bead should approach a single minimum rather than a minima?

Pg 3 line 92, add a comma after “force,”

Pg 3 line 102-107, split sentence, this is too long.

Water contact angles? Should specify these angles are water contact rather than for the oil or surfactant laden oil. Calculated, observed at the droplet contact, or measured separately with a goniometer? (e.g. pg 4 line 191, 197).

Coupling Fidelity Enhancement of DBDR Section

There is still no mention of how stirring or agitating a column during reagent coupling would mitigate some (but definitely not all) of the variability in coupling observed in a packed column reaction. Excess reagents are still usually required even when stirred, and definitely larger volumes of these reagents too, and much longer contact times, so these are still all significant advantages of the DBDR method. This is probably because DBDR is precise in delivering its solid support to the reagents for the coupling, as you do mention. However, you should also mention that it is possible to agitate or mix solid support beads in order to allow better access of the reagents to the supported reaction ligand to achieve higher yields of a completed reaction, but this will still not be as precise as DBDR for the above reasons.

Alshanski et al (2018) do state that shaking leads to limited improvement in solid supported reactions for peptide synthesis, but this same publication also states that mechanical mixing does lead to significant improvements "High rpm stirring allows for fast mixing of reagents and contributes significantly to the diffusion of reactants toward each other and the homogeneous distribution of the product and side products... [but] the use of shaking for SPPS is still preferred because of the risk of destroying the solid support when more brutal mixing methods are applied." In this cited publication, the authors optimize their mechanical mixing regime, and show it does not damage the solid support beads. You can't argue that adding stirring to a synthesis process is too complicated, as the method you are advocating for in your publication would involve a significantly larger amount of process reengineering to add single-droplet fluidics, bead and droplet manipulation, multiple coupling steps, and the as yet undefined materials retrieval / cleavage processes to your synthesis process. Also, as all of these are currently manually controlled rather than automated, automation of these processes would also be required in order to use DBDR for these applications.

I am only asking that you state that stirring or mixing could improve these existing processes, but they are not commonly used, would also require some process engineering to add them into the synthesis, and still require a much larger volume and excess ratio of reagents than employed in DBDR, and despite this, still do not achieve the efficiencies you demonstrate with DBDR. As such, you have compared your reaction to a static column slurry, as this the commonly used method for the reactions you are exploring.

You should also state that DBDR has disadvantages, mainly in that the standard resin column methods are higher throughput, so you would get more product, and that, as yet, this work has demonstrated a single coupling, so the next stage should be to multiplex this process, or to allow multiple high yield coupling reactions, and also to retrieve the reacted molecules. It would be better to discuss these aspects in the discussion rather than in the conclusions, as you already have a long description of the column process here, and this would make it more representative. Or if discussed in the conclusions and future outlook, these are the things you are going to work on in the future.

Conclusion, line 304-306 can be more concise, e.g.

"In conclusion, we highlight the major achievements of our work. Our primary physical innovation which forms the core of this manuscript is the encapsulation and ejection of microbeads from microdroplets." Could make more impact as

"In conclusion, we have demonstrated that microbeads can be controllably encapsulated and then ejected from aqueous microdroplets in order to carry out high efficiency solid-supported reactions."

Figure 1d – writing is too small to easily read, and much smaller than the fonts used on the rest of this figure. – make b-d full page width, and make sure font sizes are consistent throughout, with the caption appearing below the figure.

Some of the caption for Figure 4 is obscured by the figure, so I was not able to read this.

Ref 38 – title is capitalized whereas all the others are written as sentences, please update.

Ref 59 & 60 – journal abbreviation should be italicized.

Check if country and city is required for published book references e.g. ref 68 = Kent, UK?

Supplementary information

The references in this document are numbered in a strange way – I think it is a combination of the numbers in the main manuscript (so the first reference is #48), and then additional references from the supplement are prefixed with an S? As you only list the supplementary references at the end of this document, it is more difficult for the reader to find what you are referencing and where. Please change this to have all the references for the supplement listed in the supplement and follow one numbering system for this.

As mentioned previously, I am not a mathematician, so can't comment on the applicability or not

of the mathematical sections.

The font changes in the supplement between Time New Roman (prose / text), Arial (figure captions), and Calibri (tables)– for consistency, please check this and select the most appropriate for tables, captions and main text so you use the most appropriate. In some cases the fonts change within the same section, so this does need to be addressed.

Fig. S2 – capitalize Classius-Mossotti throughout the caption (section (b)).

Fig S7 – millimeters is spelled in UK English and the rest of the manuscript is in US English.

Fig. S9-11 – either the x axis label should be capitalized to Time, or the Y axis lower case labelled as voltage supply – it would be good to be consistent with this throughout the document.

Ref S7 – check protocol for referencing a website – it is good practice to include date accessed and some other identifiers (i.e. author, last updated) as website content can be edited / changed easily.

Reviewer-1

We thank the reviewer for accepting the manuscript for publication in Nature Communication. The reviewer's comments in the initial review were instrumental in improving the manuscript.

Reviewer-3

We are thankful that the reviewer perceives our work excellent and the representation of the revised manuscript more concise. We acknowledge the reviewer's extensive comments on our initial version were instrumental in making our manuscript better and its presentation more focused on the actual achievements of our work.

Below is our point-by-point response to the reviewer's comments on the revised manuscript.

Question-1

In some places you have a space between a number and the units, and in others the space is missing. E.g. 5.5nM/m line 142 pg 3, please check for consistency in spacing between values and units (degrees symbol should not have a space as shown on line 143, but most other cases should).

Our Response

Explanation

We acknowledge that our oversight led to inconsistencies in the representation of physical quantities in terms of magnitudes and units. An accurate and uniform representation is important for the readability and publication of the manuscript.

Action

We have extensively scrutinized the entire manuscript and rectified erroneous representation of physical quantities throughout.

Question-2

Pg 1 line 35 – remove “)” after, “initiator primer”

Our Response

Explanation

We acknowledge that a single ending bracket without a beginning bracket is confusing. For the specific point in the manuscript, we intend to clarify that the initiator strand can alternatively be called the primer strand as well.

Action

We have introduced an opening bracket before the word “primer”. So now it reads “(primer)”.

Question-3

Pg 2 line 46 – change “smaller” to “small” and consider specifying a size range in either volume or diameter, and / or also consider specifying the size of the traditional reaction chambers as well to make the comparison clearer.

Our Response

Explanation

We agree that the word small is the more appropriate word here as the word smaller means in comparison to something. However, there is no obvious comparison in that sentence with anything else. Also, a dimensional range would help the readers understand how small are the microdroplets that we are referring to.

Action

We have replaced the word “smaller” with “small” in Page 2 line 46. We have also incorporated actual range of diameters of microdroplets (1 μm - 500 μm in diameter) to quantify what we exactly mean by small.

Question-4

Pg 2 line 50 – “allows to dispense a single droplet...” is not great grammar, suggest changing to “where a single droplet can be dispensed...” or something similar.

Our Response

Explanation

We appreciate the reviewer’s attention to detail and accept his/her suggestion.

Action

We revised the sentence starting in Page 2 line 49 to, “Further, due to the advent of on-demand droplet generation, a single droplet can be dispensed.....”

Question-5

Pg 2 line 54 – “towards droplet reactors...” would read better as “towards developing / using droplet reactors...”

Our Response

Explanation

The reviewer makes a very valid comment here. We really appreciate the reviewer for paying close attention to our manuscript. It helps us improve the presentation of the manuscript.

Action

Modified the phrase in Page 2 line 54 to “towards developing droplet reactors....”

Question-6

Pg 2 line 60-64 is currently 1 sentence. This would read better if the colon was a full stop followed by “This is...”

Our Response

Explanation

We appreciate the reviewer for proposing an alternative framing of the sentence. It will make it easier for the reader to have two smaller sentences as opposed to one long sentence.

Action

In page 2 line 61, we replaced the colon with a full stop followed by “This is”

Question-7

Pg 2 line 66 – single bead should approach a single minimum rather than a minima?

Our Response

Explanation

We agree with the suggestion of the reviewer.

Action

In page 2 line 66, changed interfacial energy minima to interfacial energy minimum.

Question-8

Pg 3 line 92, add a comma after “force,”

Our Response

Explanation

We agree that it is our oversight that we missed a comma at the location mentioned.

Action

We put a comma after “force” in Page 3 line 92.

Question-9

Pg 3 line 102-107, split sentence, this is too long.

Our Response

Explanation

We agree that a sentence spanning across 5 lines may be difficult for a reader to keep track of.

Action

We split it into two sentences in the revised manuscript.

Question-10

Water contact angles? Should specify these angles are water contact rather than for the oil or surfactant laden oil. Calculated, observed at the droplet contact, or measured separately with a goniometer? (e.g. pg 4 line 191, 197).

Our Response

Explanation

Contact angle is measured at the intersection of a droplet, a surrounding suspension medium and a surface. We agree that when so many components are involved it would be easy for the readers to get confused with the references to these systems if utmost clarity is not ensured. In this discussion, we refer to the aqueous reagent microdroplet as the droplet in the system, the streptavidin surface of the bead as the solid surface and the surrounding medium which can be either air, silicone oil or silicone oil with Span 80 as the suspension medium. As the surrounding medium changes, the contact angle also is expected to change. In Page 4 line 141, we have already stated that θ is the contact angle that the reagent droplet forms on the surface of the bead. The paragraph running from line 191 to line 204 spanning across page 5 and 6, we discuss the role of the suspension medium on the value of the contact angle.

Action

We clarify that we are discussing the contact angle that the water droplet forms on the surface of the streptavidin bead in lines 191 and 192.

Question-11

There is still no mention of how stirring or agitating a column during reagent coupling would mitigate some (but definitely not all) of the variability in coupling observed in a packed column reaction. Excess reagents are still usually required even when stirred, and definitely larger volumes of these reagents too, and much longer contact times, so these are still all significant advantages of the DBDR method. This is probably because DBDR is precise in delivering its solid support to the reagents for the coupling, as you do mention. However, you should also mention that it is possible to agitate or mix solid support beads in order to allow better access of the reagents to the supported reaction ligand to achieve higher yields of a completed reaction, but this will still not be as precise as DBDR for the above reasons.

Alshanski et al (2018) do state that shaking leads to limited improvement in solid supported reactions for peptide synthesis, but this same publication also states that mechanical mixing does lead to significant improvements “High rpm stirring allows for fast mixing of reagents and contributes significantly to the diffusion of reactants toward each other and the homogeneous distribution of the product and side products... [but] the use of shaking for SPPS is still preferred because of the risk of destroying the solid support when more brutal mixing methods are applied.” In this cited publication, the authors optimize their mechanical mixing regime, and show it does not damage the solid support beads. You can’t argue that adding stirring to a synthesis process is too complicated, as the method you are advocating for in your publication would involve a significantly larger amount of process re-engineering to add single-droplet fluidics, bead and droplet manipulation, multiple coupling steps, and the as yet undefined materials retrieval / cleavage processes to your synthesis process. Also, as all of these are currently manually controlled rather than automated, automation of these processes would also be required in order to use DBDR for these applications.

I am only asking that you state that stirring or mixing could improve these existing processes, but they are not commonly used, would also require some process engineering to add them into the synthesis, and still require a much larger volume and excess ratio of reagents than employed in DBDR, and despite this, still do not achieve the efficiencies you demonstrate with DBDR. As such, you have compared your reaction to a static column slurry, as this the commonly used method for the reactions you are exploring.

You should also state that DBDR has disadvantages, mainly in that the standard resin column methods are higher throughput, so you would get more product, and that, as yet, this work has demonstrated a single coupling, so the next stage should be to multiplex this process, or to allow multiple high yield coupling reactions, and also to retrieve the reacted molecules. It would be better to discuss these aspects in the discussion rather than in the conclusions, as you already have a long description of the column process here, and this would make it more representative. Or if discussed in the conclusions and future outlook, these are the things you are going to work on in the future.

Our Response

Explanation

We acknowledge that we addressed these concerns of the reviewer during the first review iteration only in the review response. We had not explicitly stated the disadvantages of DBDR in its current state in the manuscript which is important to accurately emphasize the achievements of the work.

Actions

In lines 297 to 313 preceding the conclusion section we discuss the following:

- Other approaches exist that employ macroscale forces to enhance mixing of solid supports with reagents in synthesis columns.
- Some of these approaches achieve high reaction fidelities while not causing any bead damage.
- However, they all consume a large amount of reagents which is disadvantageous for iterative design-build-test-learn cycles.
- DBDR uses microscopic forces to precisely interface solid supports with reagents for enhanced reaction efficiency.
- The current paper presents DBDR as a novel physical approach that achieves droplet microfluidic implementation of solid-phase synthesis while also improving reaction fidelities.
- Combining parallel droplet generation, parallel particle manipulation and automation the requirements of scale, throughput, and multistep synthesis can be addressed using the concepts presented in this paper to achieve a pragmatic platform for one-bead-one-compound libraries that can minimize reagent consumption while also achieving reaction fidelities.

Question-12

Conclusion, line 304-306 can be more concise, “In conclusion, we highlight the major achievements of our work. Our primary physical innovation which forms the core of this manuscript is the encapsulation and ejection of microbeads from microdroplets.” Could make more impact as “In conclusion, we have demonstrated that microbeads can be controllably encapsulated and then ejected from aqueous microdroplets in order to carry out high efficiency solid-supported reactions.”

Our Response

Explanation

We agree with the reviewer’s suggestion.

Action

We incorporated the suggestion of the reviewer as is.

Question-13

Figure 1d – writing is too small to easily read, and much smaller than the fonts used on the rest of this figure. – make b-d full page width, and make sure font sizes are consistent throughout, with the caption appearing below the figure.

Our Response

Explanation

We agree with the reviewer's suggestion. Readability of the figure fonts is extremely important for publication.

Action

We have rearranged the subfigures of Fig. 1. In the revised Fig. 1 panels for b and c are arranged to span the entire width of the page. Panel for part d is enlarged to stretch across the entire page as well. This makes the figure labels in panel d larger and more obvious. The figure caption is put at the end of the figure.

Question-14

Some of the caption for Figure 4 is obscured by the figure, so I was not able to read this.

Our Response

Explanation

That was an oversight on our part. We thank the reviewer for catching the mistake.

Action

We rectified this issue in the revised manuscript.

Question-15

Ref 38 – title is capitalized whereas all the others are written as sentences, please update.

Our Response

Explanation

We understand that maintaining uniform formatting is important for publication. We appreciate the reviewer's attention to details.

Action

We rectified this issue in the revised manuscript.

Question-16

Ref 59 & 60 – journal abbreviation should be italicized.

Our Response

Explanation

We understand that maintaining uniform formatting is important for publication. We appreciate the reviewer's attention to details.

Action

We rectified this issue in the revised manuscript.

Question-17

Check if country and city is required for published book references e.g. ref 68 = Kent, UK?

Our Response

Explanation

We appreciate the reviewer's attention to detail. The name of the city indeed should be followed by the country in the book reference.

Action

We added U.K. after Kent in Ref 68.

Supplementary Information

Question-1

The references in this document are numbered in a strange way – I think it is a combination of the numbers in the main manuscript (so the first reference is #48), and then additional references from the supplement are prefixed with an S? As you only list the supplementary references at the end of this document, it is more difficult for the reader to find what you are referencing and where. Please change this to have all the references for the supplement listed in the supplement and follow one numbering system for this.

Our Response

Explanation

The reviewer makes a valid point. It can be confusing to go back and forth between the different numbering formats for the references as well as look back at the main text for the references.

Action

We copied the relevant references in the main text into the supplementary information and created an independent referencing system for the supplementary information.

Question-2

The font changes in the supplement between Time New Roman (prose / text), Arial (figure captions), and Calibri (tables)– for consistency, please check this and select the most appropriate for tables, captions and main text so you use the most appropriate. In some cases the fonts change within the same section, so this does need to be addressed.

Our Response

Explanation

We agree with the reviewer that the uniformity in the style of the font is important for the presentation of the manuscript.

Action

We changed the font of all figure captions (main text, supplementary and methods sections) as well as the tables to times new roman.

Question-3

Fig. S2 – capitalize Classius-Mossotti throughout the caption (section (b)).

Our Response

Explanation

We acknowledge that it was an oversight on our part to not maintain uniformity in the representation of Classius-Mossotti factor.

Action

We capitalized Classius-Mossotti factor in the caption of Fig. S2 (section (b)).

Question-4

Fig S7 – millimeters is spelled in UK English and the rest of the manuscript is in US English.

Our Response

Explanation

We agree that it is important to maintain uniformity in the usage of U.S. English or U.K. English.

Action

We corrected that in the caption of Fig. S7.

Question-5

Fig. S9-11 – either the x axis label should be capitalized to Time, or the Y axis lower case labelled as voltage supply – it would be good to be consistent with this throughout the document.

Our Response

Explanation

We agree that it is important to maintain uniformity in the usage of U.S. English or U.K. English.

Action

We corrected that in the caption of Fig. S7.

Question-6

Ref S7 – check protocol for referencing a website – it is good practice to include date accessed and some other identifiers (i.e. author, last updated) as website content can be edited / changed easily.

Our Response

Explanation

Ref S7 which is S10 in the revised submission is a datasheet for the 1cSt silicone oil used in our research. So, we have used a citation format for a data sheet that includes the name of the company, the title of the datasheet and the datasheet number.

Action

Ref S10 has been put in the form of a datasheet including all the accessible information relevant to the datasheet.